# Age-dependent formation of TMEM106B amyloid filaments in human brains

Manuel Schweighauser[1,18], Diana Arseni[1,18], Mehtap Bacioglu[2,18], Melissa Huang[1,18], Sofia Lövestam[1,18], Yang Shi[1,18], Yang Yang[1,18], Wenjuan Zhang[1,17], Abhay Kotecha[3], Holly J. Garringer[4], Ruben Vidal[4], Grace I. Hallinan[4], Kathy L. Newell[4], Airi Tarutani[5], Shigeo Murayama[6], Masayuki Miyazaki[7], Yuko Saito[8], Mari Yoshida[9], Kazuko Hasegawa[10], Tammaryn Lashley[11], Tamas Revesz[11], Gabor G. Kovacs[12,13], John van Swieten[14], Masaki Takao[15,16], Masato Hasegawa[5], Bernardino Ghetti[4], Maria Grazia Spillantini[2], Benjamin Ryskeldi-Falcon[1], Alexey G. Murzin[1], Michel Goedert[1,19]✉ & Sjors H. W. Scheres[1,19]✉

Many age-dependent neurodegenerative diseases, such as Alzheimer's and Parkinson's, are characterized by abundant inclusions of amyloid filaments. Filamentous inclusions of the proteins tau, amyloid-β, α-synuclein and transactive response DNA-binding protein (TARDBP; also known as TDP-43) are the most common[1,2]. Here we used structure determination by cryogenic electron microscopy to show that residues 120–254 of the lysosomal type II transmembrane protein 106B (TMEM106B) also form amyloid filaments in human brains. We determined the structures of TMEM106B filaments from a number of brain regions of 22 individuals with abundant amyloid deposits, including those resulting from sporadic and inherited tauopathies, amyloid-β amyloidoses, synucleinopathies and TDP-43 proteinopathies, as well as from the frontal cortex of 3 individuals with normal neurology and no or only a few amyloid deposits. We observed three TMEM106B folds, with no clear relationships between folds and diseases. TMEM106B filaments correlated with the presence of a 29-kDa sarkosyl-insoluble fragment and globular cytoplasmic inclusions, as detected by an antibody specific to the carboxy-terminal region of TMEM106B. The identification of TMEM106B filaments in the brains of older, but not younger, individuals with normal neurology indicates that they form in an age-dependent manner.

TMEM106B is a type II transmembrane protein of 274 residues that localizes to late endosomes and lysosomes[3,4]. It is expressed ubiquitously, with the highest levels in the brain, heart, thyroid, adrenal and testis[5] (https://www.proteinatlas.org). Reminiscent of the amyloid precursor protein APP, TMEM106B is sequentially processed through ectodomain shedding, followed by intramembrane proteolysis, with possible variability in the intramembrane cleavage site. Lysosomal proteases have been implicated in the cleavage of TMEM106B in the C-terminal luminal domain, but no specific enzymes have been identified. Although the cleavage site is unknown, it has been shown indirectly to be at a position close to G127. The resulting C-terminal fragment contains five glycosylation sites at N145, N151, N164, N183 and N256. Following shedding of the ectodomain, the N-terminal fragment is

cleaved by signal peptide peptidase-like 2a (SPPL2a), possibly at two different sites around residue 106 (ref. [6]).

Genetic variation at the *TMEM106B* locus has been identified as a risk factor for frontotemporal lobar degeneration with TDP-43 inclusions (FTLD-TDP), especially for individuals with granulin (*GRN*) gene mutations[7]. The change of T185 to serine (encoded by rs3173615) has been suggested to protect against FTLD-TDP (ref. [8]), possibly because the protein with a serine is more rapidly degraded[9]. In addition, the protective effects of the noncoding variant rs1990622 have been attributed to reduced expression of TMEM106B (refs. [3,8]). Levels of TMEM106B are elevated in FTLD-TDP (ref. [10]). TMEM106B has also been reported to be involved in other diseases[3,4]. Genome-wide association studies have also implicated *TMEM106B* in age-associated phenotypes in the

[1]Medical Research Council Laboratory of Molecular Biology, Cambridge, UK. [2]Department of Clinical Neurosciences, University of Cambridge, Cambridge, UK. [3]Thermo Fisher Scientific, Eindhoven, The Netherlands. [4]Department of Pathology and Laboratory Medicine, Indiana University School of Medicine, Indianapolis, IN, USA. [5]Department of Brain and Neurosciences, Tokyo Metropolitan Institute of Medical Science, Tokyo, Japan. [6]Molecular Research Center for Children's Mental Development, United Graduate School of Child Development, University of Osaka, Osaka, Japan. [7]Department of Neurology, National Center Hospital, National Center of Neurology and Psychiatry, Tokyo, Japan. [8]Department of Neuropathology, Tokyo Metropolitan Geriatric Hospital and Institute of Gerontology, Tokyo, Japan. [9]Institute for Medical Science of Aging, Aichi Medical University, Nagakute, Japan. [10]Division of Neurology, Sagamihara National Hospital, Sagamihara, Japan. [11]Department of Neurodegenerative Disease and Queen Square Brain Bank for Neurological Disorders, UCL Queen Square Institute of Neurology, London, UK. [12]Tanz Centre for Research in Neurodegenerative Diseases and Department of Laboratory Medicine and Pathobiology, University of Toronto, Toronto, Ontario, Canada. [13]Institute of Neurology, Medical University of Vienna, Vienna, Austria. [14]Department of Neurology, Erasmus Medical Centre, Rotterdam, The Netherlands. [15]Department of Clinical Laboratory, National Center of Neurology and Psychiatry, National Center Hospital, Tokyo, Japan. [16]Department of Neurology, Mihara Memorial Hospital, Isesaki, Japan. [17]Present address: Medical Research Council Prion Unit, Institute of Prion Diseases, University College London, London, UK. [18]These authors contributed equally: Manuel Schweighauser, Diana Arseni, Mehtap Bacioglu, Melissa Huang, Sofia Lövestam, Yang Shi, Yang Yang. [19]These authors jointly supervised this work: Michel Goedert, Sjors H. W. Scheres. ✉e-mail: mg@mrc-lmb.cam.ac.uk; scheres@mrc-lmb.cam.ac.uk

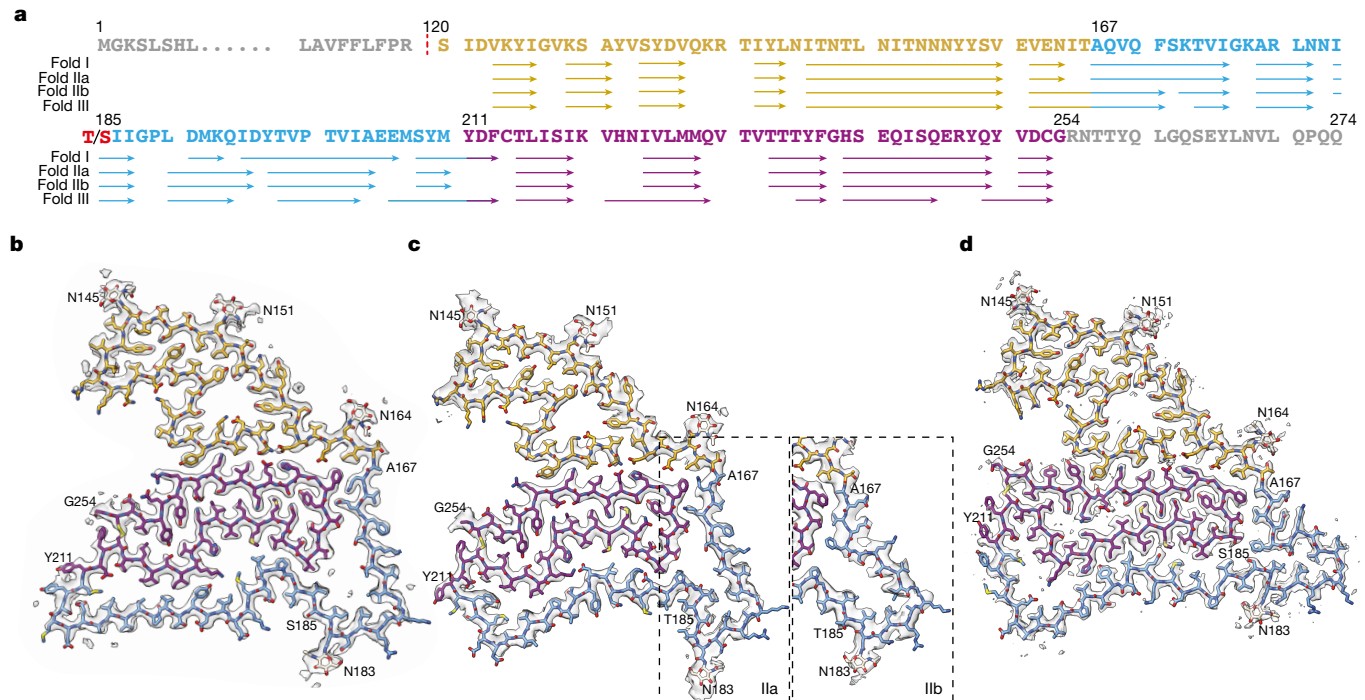

**Fig. 1 | Three TMEM106B protofilament folds from human brains. a**, Amino acid sequence of TMEM106B, with residues that form β-strands in folds I, IIa, IIb and III indicated with arrows. Residue 185 is either threonine or serine. **b–d**, Cryo-EM density maps (in transparent grey) and atomic models for TMEM106B protofilament folds I (case 1, **b**), II (case 19, **c**) and III (case 17, **d**). Two alternative conformations of fold II (IIa and IIb) are indicated within dashed boxes. Residues 120–166 are shown in yellow; residues 167–210 in light blue and residues 211–254 in magenta.

cerebral cortex[11]. Furthermore, the variant rs1990622 has been reported to correlate with reduced neuronal degeneration during ageing, independently of disease[11,12].

Previously, cryogenic electron microscopy (cryo-EM) imaging allowed atomic structure determination of filaments of the proteins tau[13–17], α-synuclein[18], amyloid-β (Aβ)[19,20] and TDP-43 (ref. [21]) that were extracted from the brains of individuals with different neurodegenerative diseases. Cryo-EM structures revealed that distinct folds characterize different diseases. For tauopathies, this has made it possible to classify known diseases further and to identify new disease entities[17].

Cryo-EM structure determination can also be used to identify previously unknown filaments. Here we have used cryo-EM to show that residues 120–254 from the luminal domain of TMEM106B form amyloid filaments in human brains. We initially observed TMEM106B filaments in the brains of individuals with familial and sporadic tauopathies, Aβ amyloidoses, synucleinopathies and TDP-43 proteinopathies. However, the role of TMEM106B filaments in disease remains unclear. They were not observed in brains from young individuals, but their presence in brains from older individuals with normal neurology (controls) indicates that TMEM106B filaments may form in an age-dependent manner. It remains to be determined how these findings relate to those from genetic association studies.

Using sarkosyl extraction protocols that were originally developed for α-synuclein[18,22], we observed a common type of filament that seemed to lack a fuzzy coat in the cryo-EM micrographs from cases of various conditions with abundant filamentous amyloid deposits. Structure determination to resolutions sufficient for de novo atomic modelling revealed that the ordered cores of these filaments consisted of residues 120–254 from the carboxy-terminal, luminal domain of TMEM106B and that the filaments were polymorphic (Fig. 1). We solved the structures of TMEM106B filaments from a number of brain regions of 22 individuals with abundant amyloid deposits, and from the frontal cortex of 3 individuals with normal neurology and no or only a few amyloid deposits (cases 1–25; Methods, Table 1 and Extended Data Table 1).

The neurodegenerative conditions for which we solved structures of associated TMEM106B filaments included sporadic and inherited Alzheimer's disease, pathological ageing, corticobasal degeneration, sporadic and inherited FTLD (FTLD-TDP types A and C, and familial frontotemporal dementia and parkinsonism linked to chromosome 17 caused by *MAPT* mutations), argyrophilic grain disease, limbic-predominant neuronal inclusion body four-repeat tauopathy, ageing-related tau astrogliopathy, sporadic and inherited Parkinson's disease, dementia with Lewy bodies, multiple system atrophy (MSA) and amyotrophic lateral sclerosis. We observed three different TMEM106B protofilament folds (I–III; Fig. 1 and Extended Data Figs. 1–4). Filaments with fold I were more common than filaments with folds II or III. For all three folds, we determined the structures of filaments that were made of a single protofilament. We also determined the structures of filaments comprising two protofilaments of fold I, related by $C_2$ symmetry. In each individual, we observed only filaments with a single fold, without a clear relationship between folds and diseases.

The TMEM106B folds shared a similar five-layered ordered core comprising residues S120–G254 and contained 17 β-strands, each ranging between 3 and 15 residues. Our best maps for filaments with folds I, II and III had resolutions of 2.6, 3.4 and 2.8 Å, and came from case 1 (sporadic Alzheimer's disease), case 19 (MSA) and case 17 (MSA), respectively (Fig. 1). TMEM106B remained fully glycosylated in all folds, as reflected by large extra densities corresponding to glycan chains attached to the side chains of N145, N151, N164 and N183. The fifth glycosylation site at N256 is outside the ordered core, with the C-terminal 20 residues being probably disordered. We divide the sequence that forms the ordered cores of the folds into three regions according to their degree of structural conservation: the amino-terminal region (S120–T166) is conserved in all three folds; the C-terminal region (Y211–G254) is conserved only in folds I and II; and the middle region (A167–M210) varies between folds.

The N-terminal region, S120–T166, forms the first two layers of the five-layered ordered cores. It comprises one long and five short β-strands

**Table 1 | Filament types from cases 1–25**

| Case | Disease | Age (years) | T185S SNP | TMEM106B filaments | Other filaments |
|------|---------|-------------|-----------|---------------------|-----------------|
| 1 | AD | 79 | SS | I-s (21%)/I-d (5%) | Aβ (37%)/tau (37%) |
| 2 | FAD | 67 | TT | I-s (16%)/I-d (1%) | Aβ (27%)/tau (56%) |
| 3 | EOAD | 58 | TT | I-s (31%)/I-d (<1%) | Aβ (15%)/tau (54%) |
| 4 | PA | 59 | TS | I-s (23%)/I-d (1%) | Aβ (76%) |
| 5 | CBD | 74 | TS | I-s (6%)/I-d (1%) | Tau (93%) |
| 6 | CBD | 79 | TS | I-s (11%)/I-d (1%) | Tau (88%) |
| 7 | FTDP-17T | 55 | TT | I-s (23%)/I-d (3%) | Tau (74%) |
| 8 | AGD | 85 | TS | III-s (8%) | Tau (92%) |
| 9 | AGD | 90 | TT | I-s (29%)/I-d (3%) | Tau (68%) |
| 10 | LNT | 66 | TT | I-s (17%)/I-d (2%) | Tau (81%) |
| 11 | ARTAG | 85 | SS | III-s (67%) | Tau (11%)/Aβ (22%) |
| 12 | PD | 87 | SS | III-s (13%) | Unknown (45%)/tau (42%) |
| 13 | PDD | 64 | TT | I-s (50%)/I-d (6%) | Aβ (28%)/unknown (16%) |
| 14 | FPD | 67 | SS | III-s (4%) | Unknown (96%) |
| 15 | DLB | 74 | SS | III-s (36%) | Unknown (64%) |
| 16 | DLB | 73 | TS | I-s (30%)/I-d (1%) | Aβ (62%)/unknown (7%) |
| 17 | MSA | 85 | SS | III-s (27%)/III-d (<1%) | αS (73%) |
| 18 | MSA | 70 | TS | I-s (13%)/I-d (5%) | αS (82%) |
| 19 | MSA | 68 | TT | IIa-s (11%)/IIb-s (4%)/II-d (<1%) | αS (85%) |
| 20 | FTLD-TDP-A | 66 | TS | I-s (21%)/I-d (30%) | Aβ (46%)/unknown (3%) |
| 21 | FTLD-TDP-C | 65 | SS | III-s (77%) | Unknown (23%) |
| 22 | ALS-TDP-B | 63 | SS | III-s (46%)/III-d (10%) | Unknown (24%)/Aβ (19%) |
| 23 | Control | 75 | TS | I-s (83%)/I-d (17%) | Undefined (<1%) |
| 24 | Control | 84 | TS | I-s (67%)/I-d (33%) | Undefined (<1%) |
| 25 | Control | 101 | TT | I-s (92%)/I-d (8%) | Undefined (<1%) |

TMEM106B filaments are indicated according to their protofilament fold (I–III) and whether they comprise one (-s) or two (-d) protofilaments. Percentages of protein filament types were calculated on the basis of the number of extracted segments from manually picked filaments (and in some cases on the number of segments after 2D classification to separate TMEM106B filaments comprising one or two protofilaments). These values may not reflect what is present in the brain, nor be directly comparable between cases. αS, α-synuclein; AD, Alzheimer's disease; AGD, argyrophilic grain disease; ALS-TDP-B, amyotrophic lateral sclerosis with TDP-43 inclusions type B; ARTAG, ageing-related tau astrogliopathy; CBD, corticobasal degeneration; control, individual with normal neurology; DLB, dementia with Lewy bodies; EOAD, early-onset Alzheimer's disease; FAD, familial Alzheimer's disease; FPD, familial PD; FTDP-17T, familial frontotemporal dementia and parkinsonism linked to chromosome 17 caused by *MAPT* mutations; FTLD-TDP-A, familial frontotemporal lobar degeneration with TDP-43 inclusions type A; FTLD-TDP-C, FTLD with TDP-43 inclusions type C; LNT, limbic-predominant neuronal inclusion body 4R tauopathy; MSA, multiple system atrophy; PA, pathological ageing; PD, Parkinson's disease; PDD, PD dementia; SNP, single nucleotide polymorphism.

that constitute a tightly packed core with hydrophobic and neutral polar residues on one side, and a large polar cavity that is filled by solvent on the other side. The three glycosylation sites in this region are located in the outer layer, adopting an extended conformation. The N-terminal residue S120 in the inner layer is buried inside the ordered core, where it packs closely against E161 from the N-terminal region and H239 and E241 from the C-terminal region (Extended Data Fig. 4).

The C-terminal region, Y211–G254, forms the two central layers of the ordered cores. It adopts a compact hairpin-like structure, the ends of which are held together by a disulfide bond between C214 and C253. Segment F237–E246 that packs against the N-terminal region has the same conformation in all three folds, whereas in the rest of the hairpin-like structure, 15 residues have opposite 'inward/outward' orientations in fold III compared with those in folds I and II. Moreover, despite similar interfaces between N- and C-terminal parts in all three folds, these regions in fold III are separated along the filament axis by one more rung than in folds I and II (Extended Data Fig. 3e).

The middle region, A167–M210, forms the fifth layer of the ordered cores and contains the fourth glycosylation site at N183. In fold I, this region packs loosely against the other side of the C-terminal hairpin-like region with the formation of three large amphipathic cavities. In fold II, these internal cavities are smaller than in fold I. We observed two sub-types of fold II (IIa and IIb) that differed mainly by the conformation

of segment A167–I187. The packing of the middle region against the C-terminal region is tightest in fold III, leaving only one sizable cavity with a salt bridge between E206 and K220. Only fold III shows *cis* isomerization of P189. In folds IIb and III, there is a large extra density at the end of the side chain of K178, suggesting that this residue may be post-translationally modified. Likewise, there is an extra density in front of the side chain of Y209 in fold I, but not in the other folds (Extended Data Fig. 1). It is possible that these residues determine the formation of the different folds. Genotyping of all individuals (Table 1) showed that the alleles encoding T185 or S185 were equally represented. Individuals with fold I were homozygous for T185 or S185, or heterozygous, indicating that fold I can accommodate a threonine or a serine at position 185. Owing to the compatibility of both residues with the glycosylation motif at N183, no differences in the associated glycan densities were observed. Fold II was found only in case 19, which was homozygous for T185. Seven out of eight individuals with fold III were homozygous for S185, with the remaining individual being heterozygous. It is possible that the packing of the side chain of residue 185 in the interior of fold III leaves insufficient space to accommodate a threonine.

In all three folds, residues G177–N183 adopt a conserved conformation, with the positively charged residues K178 and R180 pointing outwards. In filaments made of two protofilaments with fold I, two pairs of these residues are on opposite sides of a contiguous extra density that runs

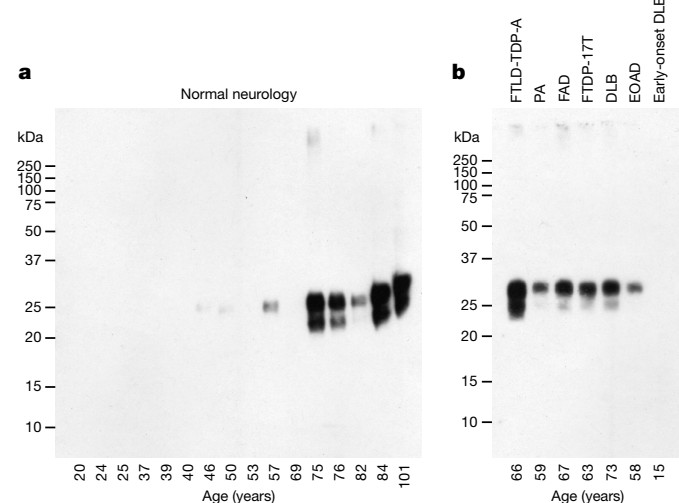

**Fig. 2 | Immunoblotting of TMEM106B inclusions from human brains.**
**a**, Analysis with anti-TMEM239 antibody (residues 239–250) of sarkosyl-insoluble extracts from the frontal cortex of 16 neurologically normal individuals aged 20–101 years. Cryo-EM structures of TMEM106B filaments were determined from the frontal cortex of individuals aged 75, 84 and 101 years. **b**, Analysis with anti-TMEM239 antibody of sarkosyl-insoluble extracts from frontal or temporal cortex of 7 individuals with abundant filamentous amyloid deposits made of various proteins. For source images for the gels, see Supplementary Fig. 1.

along the helical symmetry axis. As the cofactor responsible for this density probably does not obey the imposed helical symmetry, the map in this region is of insufficient quality to allow its identification. Although we did not solve the structures of filaments comprising two protofilaments of folds II or III, the micrographs of case 19, the only individual for which we observed filaments with fold IIa/b, and the micrographs of case 21, with fold III, also contained wider filaments that probably comprised two TMEM106B protofilaments (Extended Data Fig. 5).

In the absence of an experimentally determined native structure, we examined the structure of TMEM106B as predicted by AlphaFold[23] (Extended Data Fig. 6). Whereas the formation of amyloid filaments is often associated with natively unfolded proteins or low-complexity protein domains, the sequence S120–G254, which spans the ordered core of TMEM106B filaments, is confidently predicted to be a globular domain of the immunoglobulin-like β-sandwich fold. Glycosylation sites at N145, N151, N164 and N183 are positioned on the outside of the fold, and the disulfide bond between C214 and C253 is also predicted to form in the native structure. The β-sandwich domain is connected to a single transmembrane helix, without a flexible linker sequence. Moreover, there is a hydrophobic surface patch at this end of the domain, suggesting that it is positioned close to the membrane. It thus seems unlikely that the cleavage site at S120, the buried N-terminal residue in all TMEM106B filaments, can be accessed by lysosomal proteases. Shedding of the luminal domain may happen in a noncanonical way.

We previously showed that distinct amyloid folds of tau, α-synuclein, Aβ and TDP-43 characterize different neurodegenerative diseases[13–18,20,21]. We now describe the presence of TMEM106B filaments in many of these diseases, without a correlation between folds and diseases. Therefore, we also examined 16 brains from individuals with normal neurology that varied in age between 20 and 101 years. By immunoblotting with an antibody raised to a peptide corresponding to residues 239–250 of human TMEM106B (antibody TMEM239), the sarkosyl-insoluble fractions from disease cases showed a band of 29 kDa, which probably corresponded to the 17-kDa C-terminal fragment plus 12 kDa of glycosylation and other modifications (Fig. 2 and Extended Data Fig. 7). This band was not present in the brains from individuals with normal neurology aged less than 46 years, excluding the possibility that TMEM106B assembly was an artefact caused by tissue extraction. However, we consistently observed the 29-kDa band in the brains from control individuals older than 69 years. Interestingly, the 29-kDa band was not present in the frontal cortex from a 15-year-old individual with early-onset dementia with Lewy bodies[24] (Fig. 2b). In agreement with these observations, immunohistochemistry of brain sections with the antibody TMEM239 showed staining of inclusions in disease cases and older control individuals, but not in younger controls (Fig. 3 and Extended Data Fig. 8). It is not known how these inclusions relate to lysosomes. Cryo-EM structure determination showed the presence of TMEM106B filaments with one or two protofilaments of fold I in the frontal cortex from three controls, aged 75, 84 and 101 years.

Our results suggest that amyloid filaments of the lysosomal protein TMEM106B form in an age-dependent manner in human brains, without

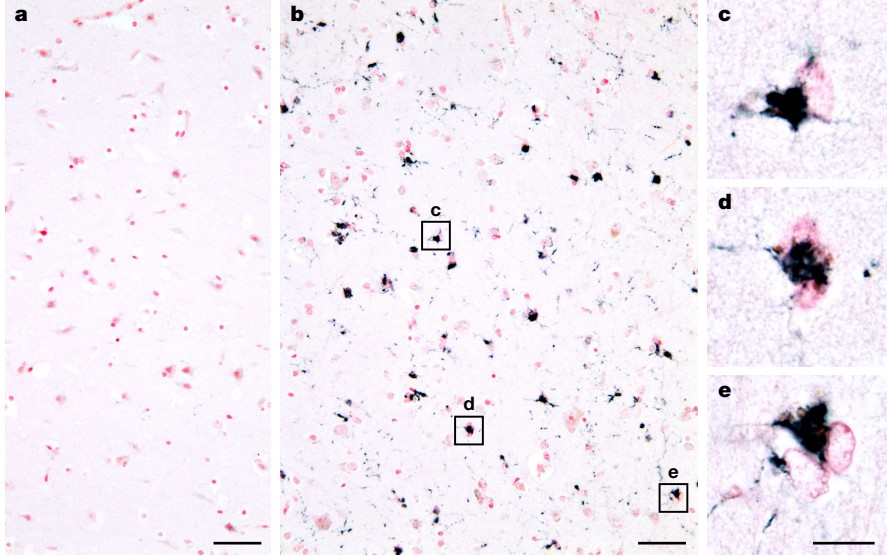

**Fig. 3 | Immunostaining of TMEM106B inclusions in human brain sections.**
**a**, **b**, Analysis with anti-TMEM239 (residues 239–250) of frontal cortex from a 25-year-old (**a**) and an 84-year-old (**b**) individual with normal neurology. No specific staining was observed in **a**, but abundant globular cytoplasmic inclusions and stained brain cell processes were present in **b**. **c**–**e**, Higher magnifications of inclusions from **b**. Nuclei were counterstained in red. Scale bars, 50 μm (**a**, **b**) and 20 μm (**c**–**e**).

a clear mechanistic connection to disease. Until now, the presence of abundant intraneuronal amyloid filaments in human tissues has always been associated with disease. Dominantly inherited mutations in the genes encoding tau, α-synuclein and TDP-43 cause neurodegenerative diseases. In addition, cryo-EM structures of amyloid filaments made of these proteins exhibit distinct folds that are characteristic of different diseases[13–18,21]. Although TMEM106B has been associated with frontotemporal dementias and other diseases, the evidence for a causal relationship between TMEM106B aggregation and disease remains unclear, and distinct TMEM106B folds do not characterize different diseases. Instead, our observations suggest that TMEM106B filaments form in an age-dependent manner. Like lipofuscin, a lysosomal complex of oxidized proteins and lipids that develops in an age-dependent manner in many tissues[25], TMEM106B filaments may also form in lysosomes, even though staining for TMEM106B inclusions was not always associated with the presence of lipofuscin autofluorescence. Lysosomal dysfunction has been implicated in the pathogenesis of neurodegenerative diseases[26]. Further studies are needed to determine whether TMEM106B filaments can be found in tissues other than the central nervous system and to assess the role of filament formation in relation to human ageing and pathologies.

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

## Methods

### Clinical history and neuropathology

We determined the cryo-EM structures of TMEM106B filaments from the brains of 25 individuals (Table 1 and Extended Data Table 1). Most individuals have been reported previously[14,16–18,20]. Unpublished cases are described below. Early-onset Alzheimer's disease (EOAD; case 3) was in a 58-year-old woman who died with a neuropathologically confirmed diagnosis following a 7-year history of memory loss. FTDP-17T (case 7) was in a 55-year-old man who died with a neuropathologically confirmed diagnosis following a 2-year history of behavioural changes, aphasia and dementia caused by a P301L substitution in *MAPT*. His brother, sister and mother were also affected. Sporadic PD (case 12) was in an 87-year-old male who died with a neuropathologically confirmed diagnosis following an 8-year history of PD. Inherited PD (case 14) was in a 67-year-old woman who died with a neuropathologically confirmed diagnosis following a 10-year history of PD caused by a G51D substitution in *SNCA*. FTLD-TDP-C (case 21) was in a 65-year-old woman who died with a neuropathologically confirmed diagnosis following a 9-year history of semantic dementia. ALS (case 22) was in a 63-year-old woman who died with a neuropathologically confirmed diagnosis of ALS stage 4, type B TDP-43 pathology, following a history of 2 years and 5 months of motor symptoms, without dementia. Control 1 (case 23) was a 75-year-old man who died of coronary heart disease without neuropathological abnormalities. Control 2 (case 24) was a 84-year-old man with mild tau pathology (Braak stage 1) who died of sepsis. Control 3 (case 25) was a 101-year-old man with mild tau pathology (Braak stage 1) and mild cerebral amyloid angiopathy who died of pneumonia.

### Extraction of TMEM106B filaments

Sarkosyl-insoluble material was extracted from frontal cortex (EOAD, FTLD-TDP-C and control cases 1–16), cingulate cortex (sporadic PD), temporal cortex (inherited PD and FTDP-17T) and motor cortex (ALS), essentially as described previously[22]. Similar extraction methods were used for all other cases, which have been described in the references in Extended Data Table 1. The original sarkosyl extraction method, which we used in our work on the cryo-EM structures of tau filaments from Alzheimer's disease, chronic traumatic encephalopathy and Pick's disease[13–15], uses sarkosyl only after the first, low-speed centrifugation step[27]. A previously published method[22] also uses sarkosyl at the beginning (before the first centrifugation step). This protocol change was essential for detecting abundant TMEM106B filaments, possibly because clumped filaments end up in the first pellet when sarkosyl is not yet present in the original method. In addition, the previously published method[22] uses a gentler clearing spin at the end, which results in an increase in the amount of filaments in the final sample. In brief, tissues were homogenized in 20 vol (w/v) extraction buffer consisting of 10 mM Tris-HCl, pH 7.4, 0.8 M NaCl, 10% sucrose and 1 mM EGTA. Homogenates were brought to 2% sarkosyl and incubated for 30 min at 37 °C. Following a 10-min centrifugation at 10,000$g$, the supernatants were spun at 100,000$g$ for 20 min. The pellets were resuspended in 700 µl g$^{-1}$ extraction buffer and centrifuged at 5,000$g$ for 5 min. The supernatants were diluted threefold in 50 mM Tris-HCl, pH 7.4, containing 0.15 M NaCl, 10% sucrose and 0.2% sarkosyl, and spun at 166,000$g$ for 30 min. Sarkosyl-insoluble pellets were resuspended in 50 µl g$^{-1}$ of 20 mM Tris-HCl, pH 7.4 containing 100 mM NaCl.

### Immunoblotting and immunohistochemistry

Immunoblotting was carried out as described previously[28]. Sarkosyl-insoluble pellets were diluted 1:3 and sonicated in a water-bath for 10 min at 50% amplitude (QSonica). They were resolved on 12% Bis-Tris gels (Novex) and the antibody TMEM239 (a rabbit polyclonal antibody that was raised to a synthetic peptide corresponding to residues 239–250 of human TMEM106B) was used at 1:2,000. To enhance the signal, membranes were boiled in PBS for 10 min at 95 °C. For immunohistochemistry, formalin-fixed, paraffin-embedded 8-µm-thick sections were incubated overnight in xylene. Following deparaffinization, the sections underwent heat-induced epitope retrieval in Tris-EDTA buffer (10 mM Tris base, 1 mM EDTA, 0.05% Tween 20, pH 9). Peroxidase was quenched by incubation in 3% hydrogen peroxide in PBS containing 20% methanol for 30 min, followed by a 15-min incubation in BLOXALL endogenous blocking solution (Vector Laboratories). After a brief wash in PBS + 0.3% Triton X-100 (PBST), the sections were incubated in blocking buffer (2.5% bovine serum albumin, 5% horse serum in PBST) for 1 h at room temperature. This was followed by an overnight incubation at 4 °C with primary antibody in blocking solution (TMEM239 was used at 1:500 and N-terminal rabbit polyclonal TMEM106B antibody A303-439A (Bethyl Laboratories)[29], which was raised to a synthetic peptide corresponding to residues 1–50 of human TMEM106B, was used at 1:250). After three washes with PBST, the sections were incubated with ImmPRESS-HRP polymer anti-rabbit detection antibody (Vector Laboratories) for 2 h at room temperature. After another three washes with PBST, Vector SG substrate (peroxidase) was added to visualize the antigen. Sections were counterstained with nuclear fast red and covered with a coverslip using Entellan mounting medium (Merck). Images were acquired with a QImaging Retiga 2000R CCD camera using an Olympus BX50 microscope.

### Cloning

TMEM106B C-terminal fragment (120–274) incorporated in pET3A was purchased from Genscript. The construct lacking residues 239–250 (Δ239–250) was made using in vivo assembly[30]. Forward and reverse primers were obtained from Integrated DNA Technologies and were designed to share 15–20 nucleotides of homologous region and 15–30 nucleotides for annealing to the template, flanking the region of deletion, with melting temperatures ranging from 58 to 65 °C. Before transformation, PCR products were treated with DpnI.

### Purification of recombinant TMEM106B

Plasmids were transformed into *Escherichia coli* BL21 cells (DE3pLys; Agilent). One plate was used to inoculate 500 ml terrific broth (TB), supplemented with 2.5 mM MgSO$_4$ and 2% ethanol, 100 mg l$^{-1}$ ampicillin, and the bacteria were grown with shaking at 220 r.p.m. at 37 °C, until an OD of 0.8 was reached. Expression was then induced with 1 mM IPTG, followed by growth for 4 h at 37 °C. To check for TMEM106B expression, 1 ml of induced culture was spun at 3,000$g$ for 10 s, resuspended in 50 µl gel loading buffer and used for immunoblotting. Bacterial cells expressing TMEM106B were collected by centrifugation for 20 min at 4,000$g$ at 4 °C and resuspended in cold buffer A: 4× PBS, pH 7.4, 25 mM dithiothreitol (DTT), 0.1 mM phenylmethylsulfonyl fluoride and complete protease inhibitor tablets (4 tablets per 100 ml). Resuspension was performed using a Polytron with a 10:1 volume-to-weight ratio of pellet to buffer. The homogenized pellets were sonicated (40% amplitude, 5 s on, 10 s off, for 6 min) at 4 °C. Lysed cells were then centrifuged at 30,000$g$ for 40 min at 4 °C, and the pellets were resuspended in buffer A plus 2 M urea and 2% Triton, incubated for 30 min at 40 °C and centrifuged at 30,000$g$ for 20 min at 25 °C. This resuspension step was repeated three times. Subsequently, the pellets, appearing as dense white matter indicative of inclusion bodies, were resuspended in buffer A plus 2 M urea, incubated for 30 min at 40 °C and centrifuged at 30,000$g$ for 20 min at 25 °C. Finally, the pellets were resuspended in buffer A plus 8 M urea, using a 20:1 volume-to-weight ratio, for 1 h with shaking at 100 r.p.m. at 60 °C, and centrifuged at 30,000$g$ for 20 min at 25 °C. These pellets were resuspended in 4× PBS, pH 7.4, 50 mM DTT and 8 M urea, and left shaking overnight at 100 r.p.m. at 60 °C. The resuspended pellets were centrifuged at 45,000$g$ for 30 min and the supernatants were concentrated, followed by buffer exchange using a PD10 desalting column into 2× PBS, pH 7.4 and 50 mM DTT. Samples were further concentrated to 3 mg ml$^{-1}$ using a 3-kDa-cutoff molecular weight

concentrator, and used for immunoblotting to establish the specificity of the antibody TMEM239 (Extended Data Fig. 7).

## Genotyping of the rs3173615 variant (T185S)

Genomic DNA was extracted from human brains using the DNeasy blood and tissue kit (QIAGEN). PCR amplification of a 470-bp fragment encompassing exon 6 of *TMEM106B* used GoTaq DNA Polymerase (Promega). The primers were: 5′-GGTTTAATTTTCTTTGACATTTTGG-3′ (forward) and 5′-GGCTCAAGCAGTCCACTGAG-3′ (reverse). We analysed the nucleotide variation C>G at position chr7:12,229,791 (hg38).

## Cryo-EM

For all cases, except EOAD, FTDP-17T, LNT, sporadic PD, inherited PD, FTLD-TDP-C, ALS and control cases 1–3, the cryo-EM datasets have been described in the references in Extended Data Table 1. For the remaining cases, resuspended sarkosyl-insoluble pellets were applied to glow-discharged holey carbon gold grids (Quantifoil R1.2/1.3, 300 mesh) and plunge frozen in liquid ethane using an FEI Vitrobot Mark IV. FTLD-TDP-A, FTLD-TDP-C and ALS samples were treated with 0.4 mg ml$^{-1}$ pronase for 50–60 min before glow discharging, which further improved the TMEM106B filament yield. Images for cases of EOAD, FTDP-17T, LNT, FPD, FTLD-TDP-C and ALS were acquired using EPU software on Thermo Fisher Titan Krios microscopes, operated at 300 kV, with a Gatan K2 or K3 detector in counting mode, using a Quantum energy filter (Gatan) with a slit width of 20 eV to remove inelastically scattered electrons. Images for EOAD, sporadic PD and control cases 1–3 were acquired on a Thermo Fisher Titan Krios, operated at 300 kV, using a Falcon-4 detector and no energy filter.

## Helical reconstruction

Movie frames were gain corrected, aligned, dose weighted and then summed into a single micrograph using RELION's own motion correction program[31]. The micrographs were used to estimate the contrast transfer function (CTF) using CTFFIND-4.1 (ref. [32]). All subsequent image-processing steps were performed using helical reconstruction methods in RELION (refs. [33,34]). TMEM106B filaments were picked manually, as they could be distinguished from filaments made of tau, Aβ, α-synuclein and TDP-43 by their general appearance and the apparent lack of a fuzzy coat. TMEM106B filaments comprising one or two protofilaments were picked separately. For all datasets, reference-free 2D classification was performed to select suitable segments for further processing. Initial 3D reference models were generated de novo from the 2D class averages using an estimated rise of 4.75 Å and helical twists according to the observed crossover distances of the filaments in the micrographs[31] for datasets of cases 10 (LNT; folds I-s and I-d), 18 (MSA; fold I-d), 19 (MSA; folds IIa and IIb) and 17 (MSA; fold III). Refined models from these cases, low-pass filtered to 10–20 Å, were used as initial models for the remaining cases. Combinations of 3D auto-refinements and 3D classifications were used to select the best segments for each structure. For all datasets, Bayesian polishing[35] and CTF refinement[36] were performed to further increase the resolution of the reconstructions. Final reconstructions were sharpened using the standard post-processing procedures in RELION, and overall final resolutions were estimated from Fourier shell correlations at 0.143 between the two independently refined half-maps, using phase randomization to correct for convolution effects of a generous, soft-edged solvent mask[37]. Further details of data acquisition and processing for the datasets that resulted in the best maps for five different TMEM106B filaments (filaments made of one or two protofilaments with fold I, as well as filaments made of one protofilament with fold IIa, fold IIb or fold III) are given in Extended Data Table 2.

## Model building

TMEM106B was identified by scanning the human proteome with different sequence motifs[38], deduced from initial maps of folds I and III.

A simple combination of four N-glycosylation motifs N-x-[ST] with the exact spacers, N-x-[ST]-x(3)-N-x-[ST]-x(10)-N-x-[ST]-x(16)-N-x-[ST], was the most effective, resulting in a hit for only TMEM106B, the sequence of which corresponded well to the entire maps. Atomic models comprising three β-sheet rungs were built de novo in Coot[39] in the best available map for each of the five different structures. Coordinate refinement was performed in ISOLDE (ref. [40]). Dihedral angles from the middle rung, which was set as a template in ISOLDE, were also applied to the rungs below and above. For each refined structure, separate model refinements were performed for the first half-map, after increasing the temperature to 300 K for 1 min, and the resulting model was then compared to that same half-map (FSC$_{work}$) as well as the other half-map (FSC$_{test}$) to confirm the absence of overfitting. Final statistics for the refined models are given in Extended Data Table 2.

## Ethical review processes and informed consent

Studies carried out at Indiana University, Tokyo Metropolitan Institute of Medical Science, Tokyo National Center Hospital, UCL Queen Square Institute of Neurology, Toronto University, Vienna Medical University, Rotterdam University and the Edinburgh Brain and Tissue Bank were approved through the ethical review processes at each university's Institutional Review Board (IRB). Informed consent was obtained from the patients' next of kin. This study was approved by the Cambridgeshire 2 Research Ethics Committee (09/H0308/163).

## Reporting summary

Further information on research design is available in the Nature Research Reporting Summary linked to this paper.

## Data availability

Cryo-EM maps have been deposited in the Electron Microscopy Data Bank (EMDB) under accession numbers 14174 for I-s of case 1, 14176 for I-d of case 18, 14187 for IIa-s and 14188 for IIb-s of case 19, and 14189 for III-s of case 17. Corresponding refined atomic models have been deposited in the Protein Data Bank under accession numbers 7QVC for I-s of case 1, 7QVF for I-d of case 18, 7QWG for IIa-s and 7QWL for IIb-s of case 19, and 7QWM for III-s of case 17.

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

**Acknowledgements** We thank the patients' families for donating brain tissues; U. Kuederli, M. Jacobsen, F. Epperson and R. M. Richardson for human brain collection and technical support; E. Gelpi for preparing brain samples from the ARTAG case; T. Darling and J. Grimmett for help with high-performance computing; M. Arai, T. Katsinelos and N. Obata for help with genotyping; I. Lavenir and J. A. Macdonald for helpful discussions; the EM facility of the Medical Research Council (MRC) Laboratory of Molecular Biology for help with cryo-EM data acquisition; the Edinburgh Brain and Tissue Bank, which is supported by the MRC, for providing brain samples of controls with normal neurology. We acknowledge Diamond Light Source for access and support of the cryo-EM facilities at the UK's national Electron Bio-imaging Centre (under proposals EM17434-75 and BI23268-49), funded by the Wellcome Trust, MRC and BBSRC. This work was supported by the MRC (MC_UP_120/25 to B.R.-F., MC_U105184291 to M.G., and MC_UP_A025_1013 to S.H.W.S.), the EU/EFPIA/Innovative Medicines Initiative [2] Joint Undertaking IMPRiND (project 116060, to M.G. and M.G.S.), the Japan Agency for Science and Technology (Crest, JPMJCR18H3, to M. Hasegawa.), the Japan Agency for Medical Research and Development (AMED, JP20dm0207072, to M. Hasegawa., and AMED, JP21wm0425019, to M.T.), the Japan Society for the Promotion of Science (JSPS, Kakenhi 21K06417, to M.T.), the US National Institutes of Health (P30-AG010133, UO1-NS110437 and RF1-AG071177, to R.V. and B.G.) and the Department of Pathology and Laboratory Medicine, Indiana University School of Medicine (to R.V., K.L.N. and B.G.). M.G.S. was supported by the NIHR Cambridge Biomedical Research Centre. G.G.K. was supported by the Safra Foundation and the Rossy Foundation. T.R. was supported by the National Institute for Health Research Queen Square Biomedical Research Unit in Dementia. M.T. was supported by intramural funds from the National Center of Neurology and Psychiatry. T.L. holds an Alzheimer's Research UK Senior Fellowship. The Queen Square Brain Bank is supported by the Reta Lila Weston Institute for Neurological Studies. For the purpose of Open Access, the authors have applied a CC-BY public copyright licence to any author accepted manuscript version arising from this submission.

**Author contributions** D.A., M.B., M. Huang, S.L., Y. Shi and Y.Y. are listed in alphabetical order. K.L.N., S.M., M.M., Y. Saito, M.Y., K.H., T.L., T.R., G.G.K., J.v.S., M.T., M. Hasegawa., B.G., B.R.-F. and M.G. identified patients and performed neuropathology; M.S., Y. Shi, M. Huang., D.A., Y.Y., W.Z., H.J.G., R.V., G.I.H., A.T. and M. Hasegawa. performed brain sample analysis; M.S., Y. Shi, D.A., Y.Y., W.Z. and A.K. collected cryo-EM data; M.S., Y. Shi, D.A., Y.Y., S.L., W.Z., B.R.-F., A.G.M. and S.H.W.S. analysed cryo-EM data; M.S. and M. Huang. performed immunoblot analysis; S.L. performed recombinant protein expression and epitope mapping; M.B. and M.G.S. performed immunohistochemistry. M.G. and S.H.W.S. supervised the project. All authors contributed to the writing of the manuscript.

**Competing interests** The authors declare no competing interests.

**Additional information**
**Correspondence and requests for materials** should be addressed to Michel Goedert or Sjors H. W. Scheres.

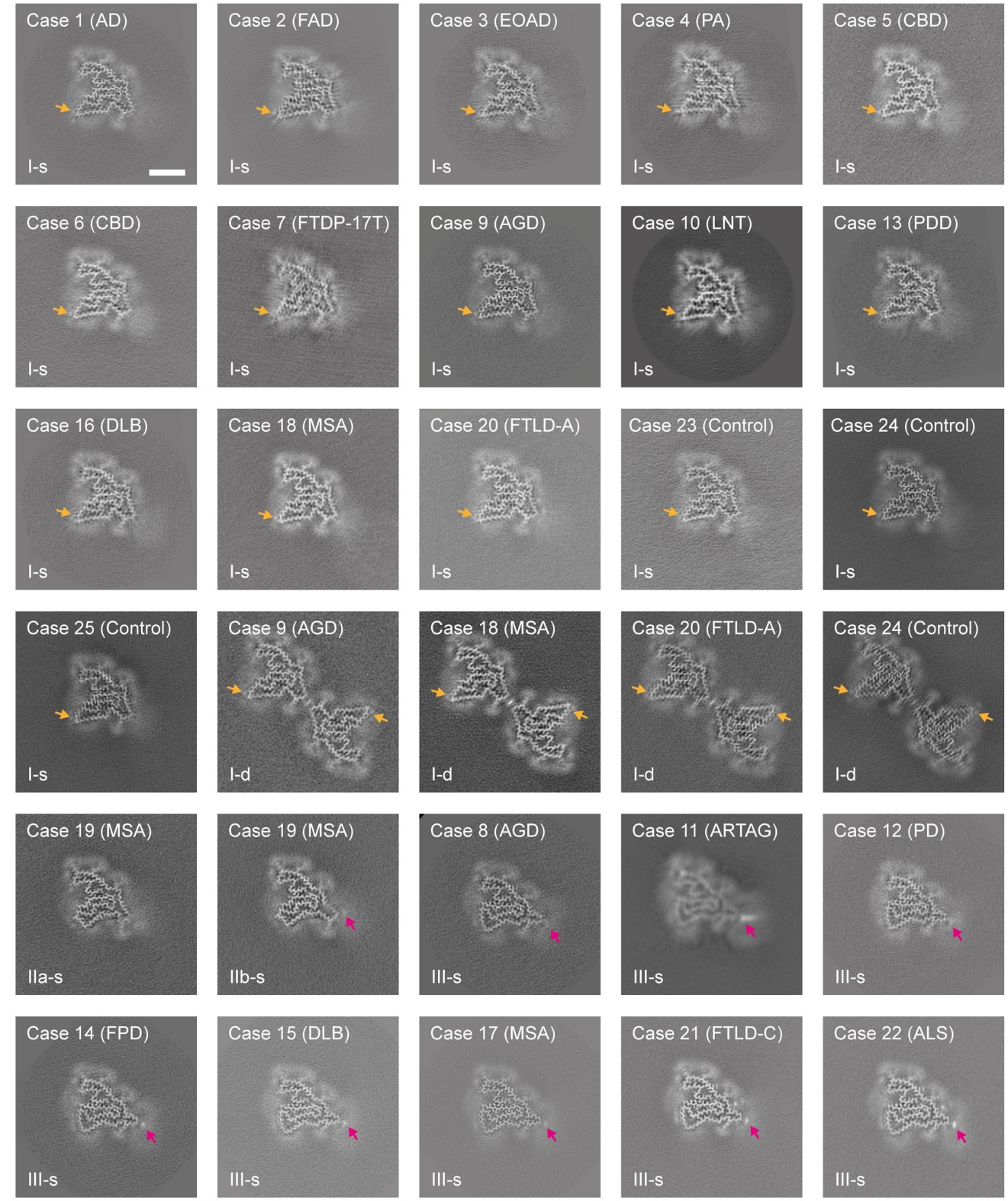

**Extended Data Fig. 1 | TMEM106B filament reconstructions.** Cross-sections of TMEM106B filaments, perpendicular to the helical axis and with a projected thickness of approximately one β-rung, for all cases examined. Orange arrows point at additional densities in front of Y209; magenta arrows point at additional densities in front of K178. Scale bar, 5 nm.

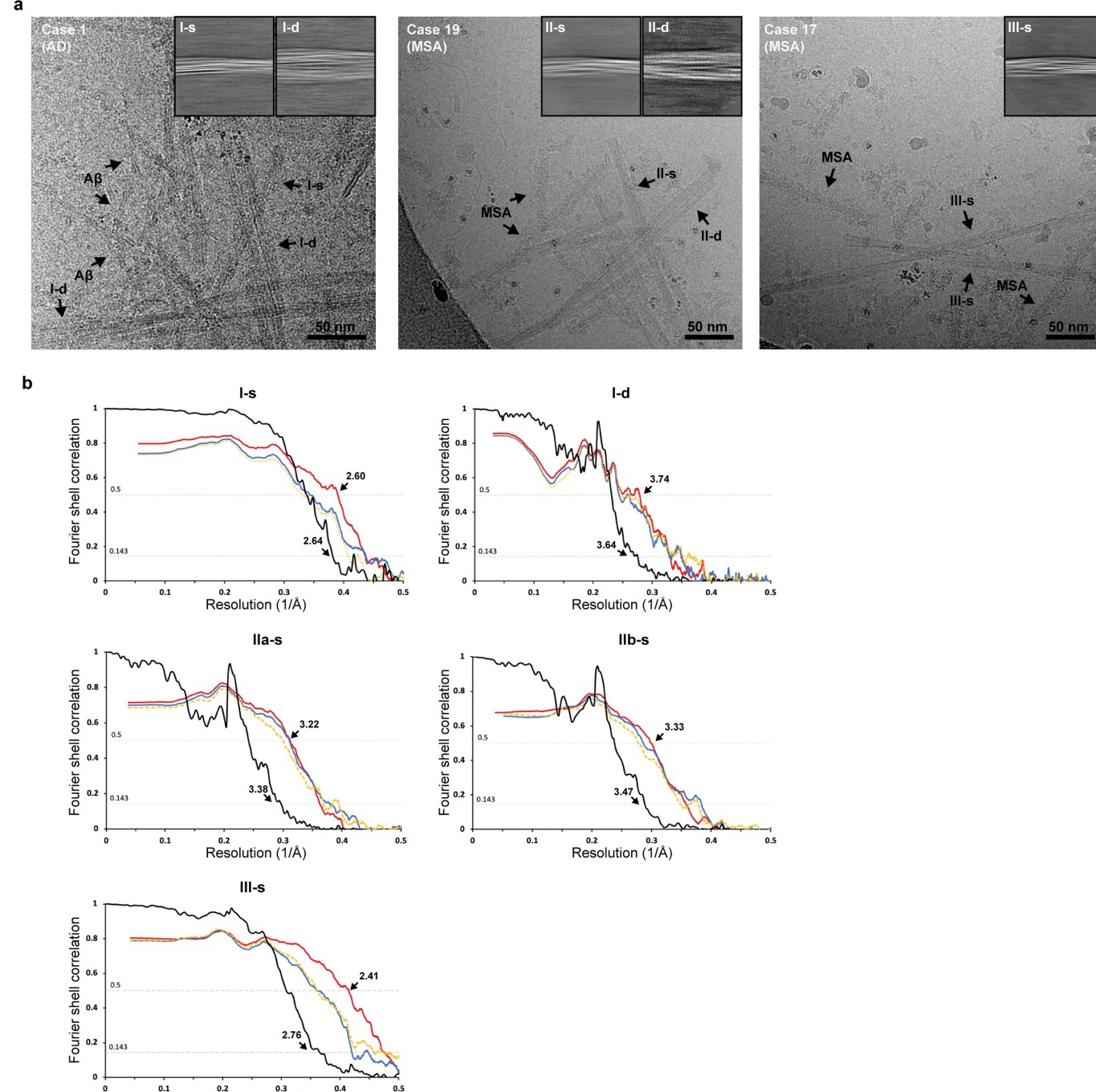

**Extended Data Fig. 2 | Cryo-EM images and resolution estimates. a**. Cryo-EM micrographs of cases 1, 19 and 17, with insets showing representative 2D class averages of TMEM106B filaments I-s, I-d, II-s, II-d and III-s. Examples of the different types of TMEM106B filaments, as well as filaments of Aβ and MSA filaments of α-synuclein, are indicated in the micrographs with black arrows. Scale bars, 50 nm. **b**. Fourier shell correlation (FSC) curves for cryo-EM maps and structures of TMEM106B filaments I-s, I-d, II-s, II-d and III-s. FSC curves for two independently refined cryo-EM half maps are shown in black; for the final refined atomic model against the final cryo-EM map in red; for the atomic model refined in the first half map against that half map in blue; and for the refined atomic model in the first half map against the other half map in yellow.

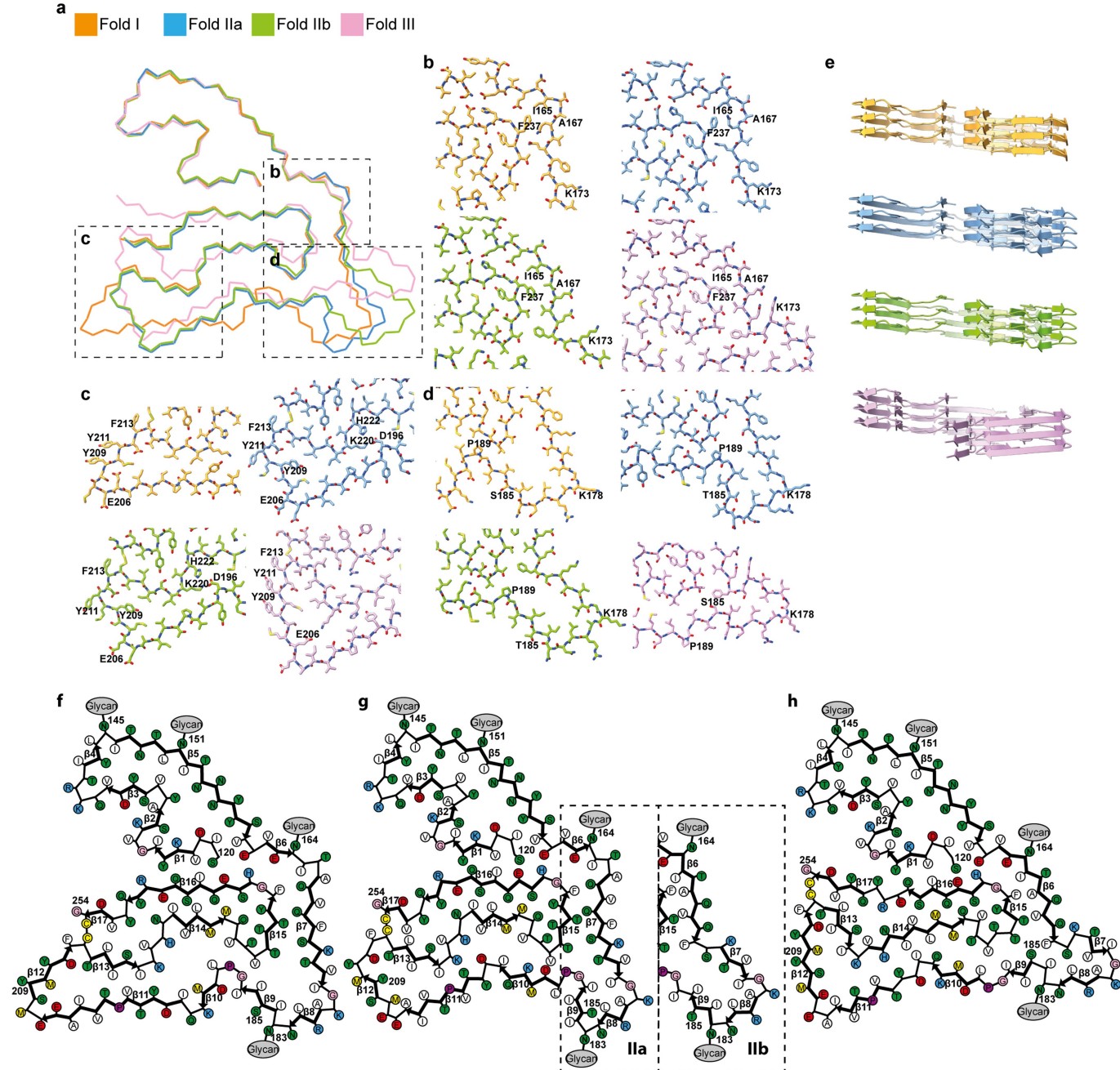

**Extended Data Fig. 3 | Comparisons of TMEM106B folds I-III. a** Ribbon view of folds I-III aligned at residues 120–166 (centre) with close-up views for three regions (**b-d**). **e.** Cartoon views for three subsequent β-rungs for each fold, viewed from the left-hand side of the ribbon view in panel A. Fold I is shown in orange; fold IIa in blue; fold IIb in green; fold III in pink. Schematics of fold I (**f**), fold II (**g**), and fold III (**h**). Negatively charged residues are shown in red, positively charged residues in blue, polar residues in green, apolar residues in white, sulfur-containing residues in yellow, prolines in purple, and glycines in pink. Thick connecting lines with arrowheads indicate β-strands.

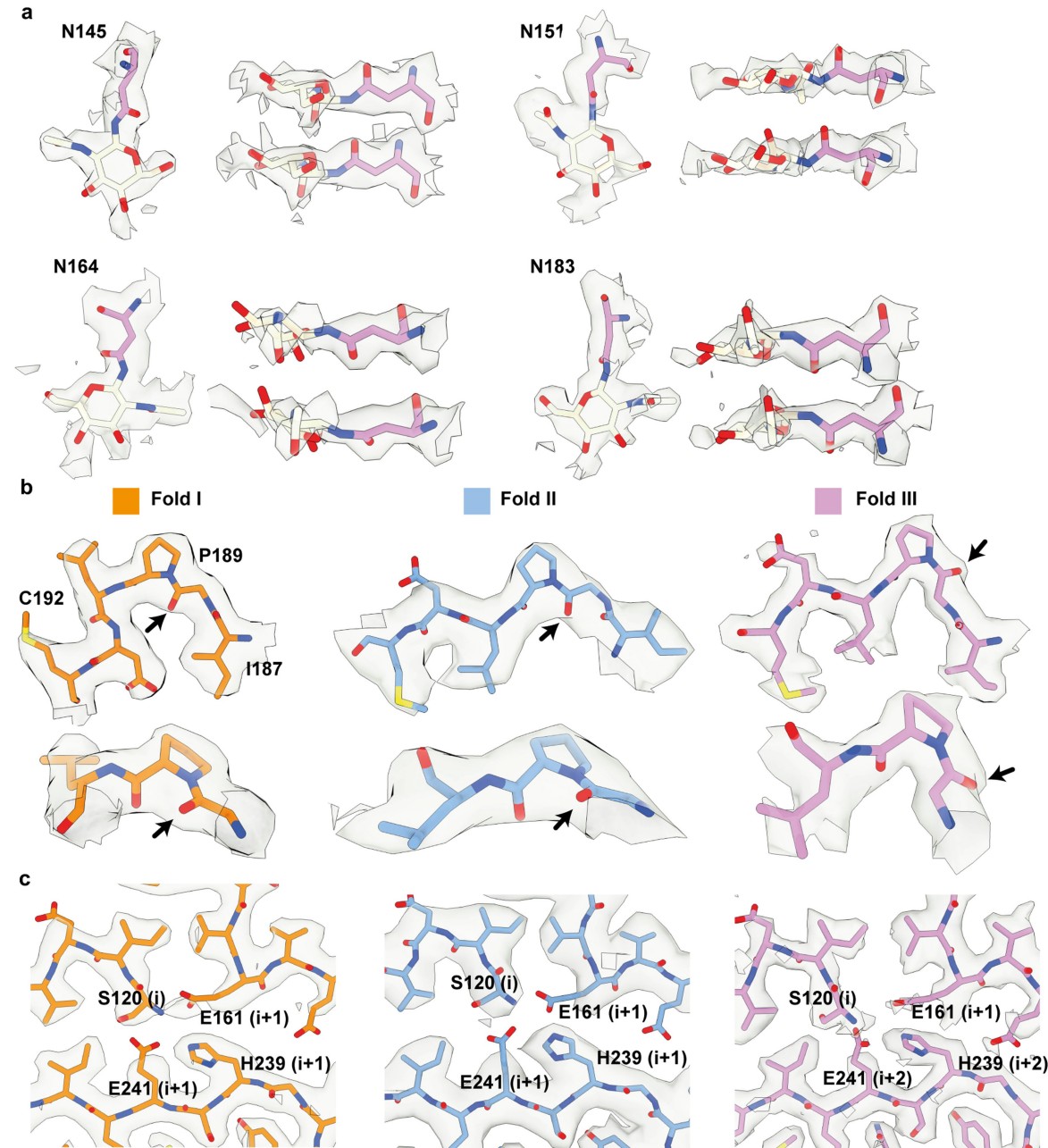

**Extended Data Fig. 4 | Close-up views of cryo-EM densities. a**. Cryo-EM densities (transparent grey) for N-linked glycosylation of asparagines 145, 151, 164 and 183 in fold III, showing the first, most ordered glcNac saccharide of the glycan chains. **b**. Density for residues 187–192 in fold I (left, orange), fold II (middle, blue) and fold III (right, pink), as viewed from the top (top panels) and the side (bottom panels). Carbonyl oxygens of P189 are indicated with black arrows. In folds I and II, P189 adopts a *trans*-configuration, whereas in fold III, P189 adopts a *cis*-conformation. **c**. Density for the N-terminal S120 residue, and its surrounding residues E161, H239, E241. The latter are one β-rung above (i+1) the β-rung of S120 (i) in folds I and II. In fold III, H239 and E241 are two β-rungs above (i+2) the β-rung of S120.

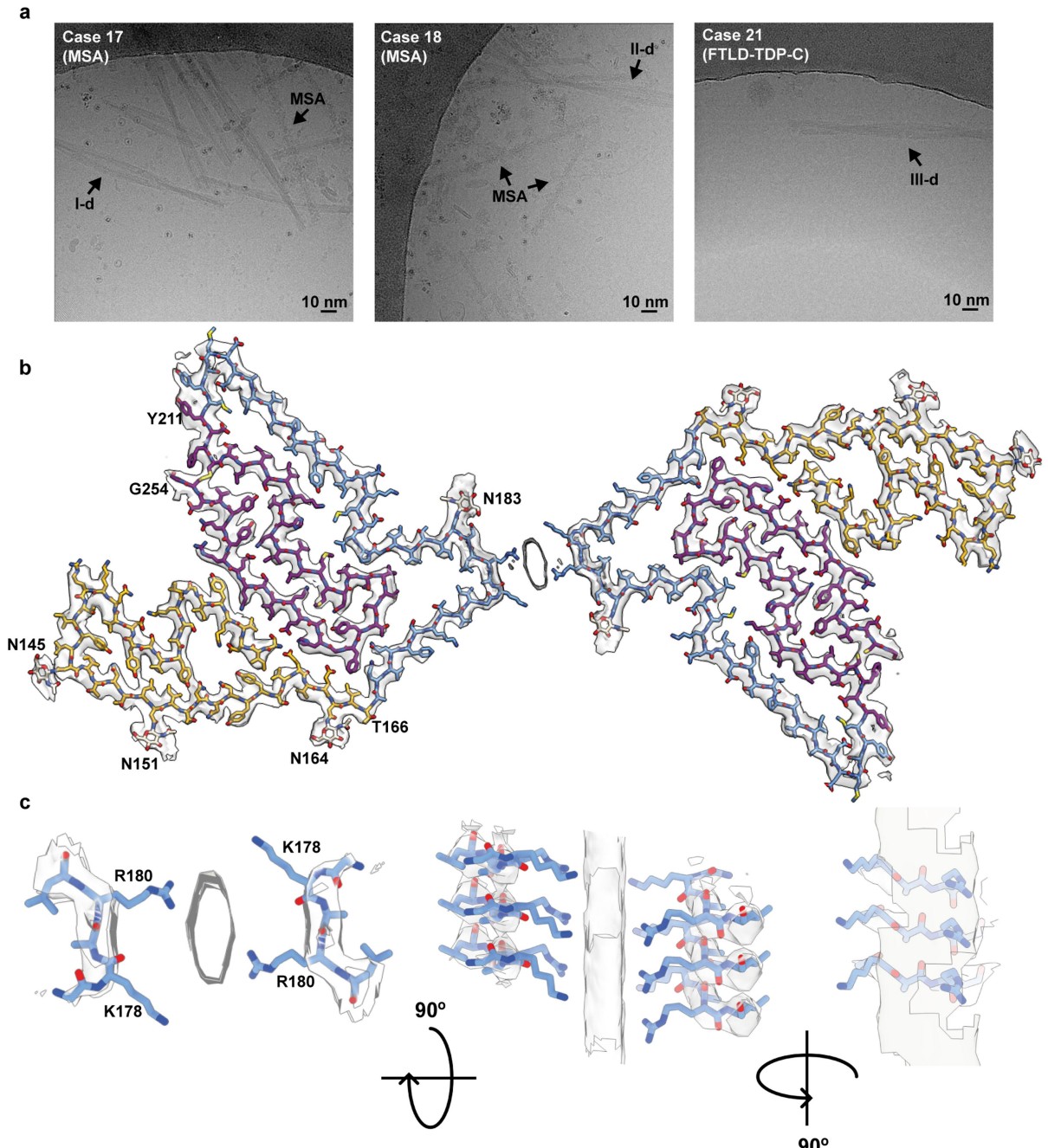

**Extended Data Fig. 5 | TMEM106B filaments comprising two protofilaments. a.** Cryo-EM micrographs with filaments comprising two protofilaments, with fold I for MSA case 18 (I-d; left), with putative fold II for MSA case 19 (II-d; middle), and with putative fold III for FTLD-TDP-C case 21 (III-d; right); α-synuclein filaments typical for MSA are also indicated (MSA). **b.** Cryo-EM density map and atomic model of TMEM106B filaments comprising two protofilaments of fold I. **c.** Three orthogonal close-up views of the inter-protofilament interface.

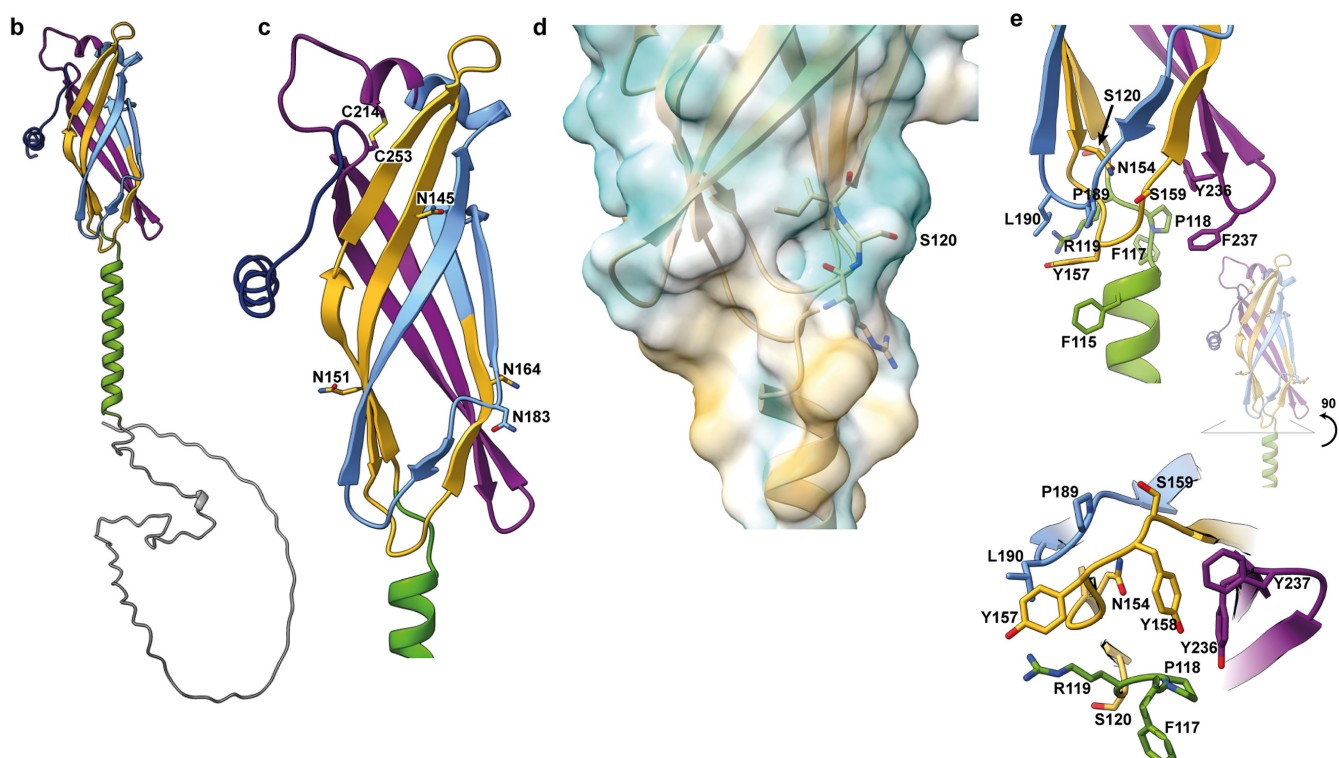

**a**

```
  1  MGKSLSHLPL  HSSKEDAYDG  VTSENMRNGL  VNSEVHNEDG  RNGDVSQFPY  VEFTGRDSVT  CPTCQGTGRI  PRGQENQLVA  LIPYSDQRLR  90

 91  PRRTKLYVMA  SVFVCLLLSG  LAVFFLFPRS  IDVKYIGVKS  AYVSYDVQKR  TIYLNITNTL  NITNNNYYSV  EVENITAQVQ  FSKTVIGKAR  180

181  LNNITIIGPL  DMKQIDYTVP  TVIAEEMSYM  YDFCTLISIK  VHNIVLMMQV  TVTTTYFGHS  EQISQERYQY  VDCGRNTTYQ  LGQSEYLNVL  270

271  QPQQ
```

**Extended Data Fig. 6 | AlphaFold prediction of TMEM106B. a**. Amino acid sequence of TMEM106B. Predicted α-helices are represented with wavy blue lines; predicted β-strands with red arrows. Residues 1–90, 91–119, 120–166, 167–210, 211–254 and 255–274 are coloured in grey, green, yellow, light blue, magenta and dark blue, respectively. **b**. AlphaFold prediction of TMEM106B. **c**. Close-up view of the AlphaFold prediction of part of the transmembrane helix (green) and the lumnal domain, with glycosylation sites N145, N151, N164, N183 and disulfide bridge C214, C253 shown as sticks. **d**. Hydrophobicity surface view at the interface between the lumenal domain and the transmembrane helix. Residues 119–121 are shown as sticks. **e**. Two orthogonal close-up views of residues close to the lysosomal membrane surface.

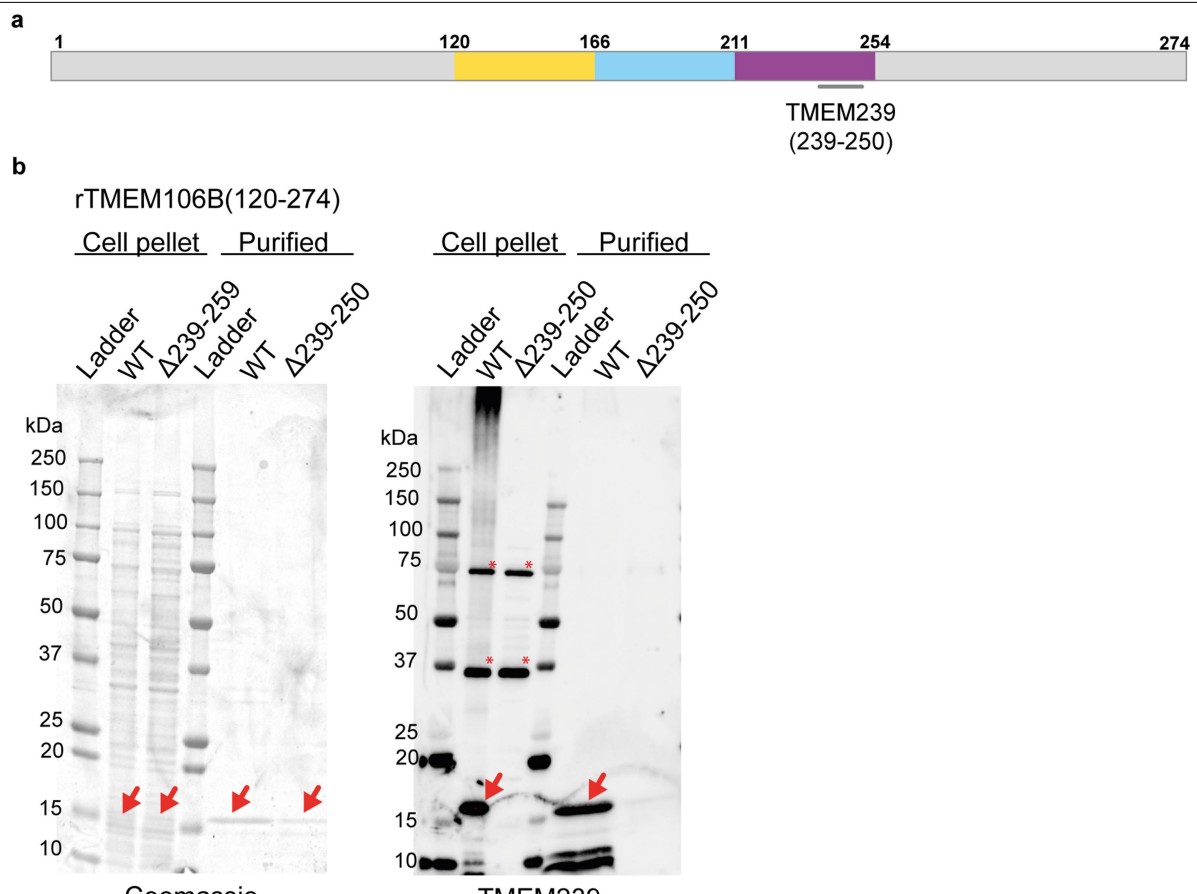

**Extended Data Fig. 7 | Immunoblot analysis of TMEM106B expressed in *E. coli* establishes the specificity of antibody TMEM239. a**. Diagram of the TMEM106B sequence, coloured in accordance with Fig. 1, with the immunogen of antibody TMEM239 (residues 239–250) indicated. **b**. Coomassie blue-stained gel and immunoblot (antibody TMEM239) of recombinant C-terminal TMEM106B fragment (WT; residues 120–274) and the immunogen-deletion construct (Δ239-250). Pellets from 1 ml cell extracts and TMEM106B purified from inclusion bodies (see Methods) were used. Red arrows indicate TMEM106B bands; red asterisks indicate non-specific binding.

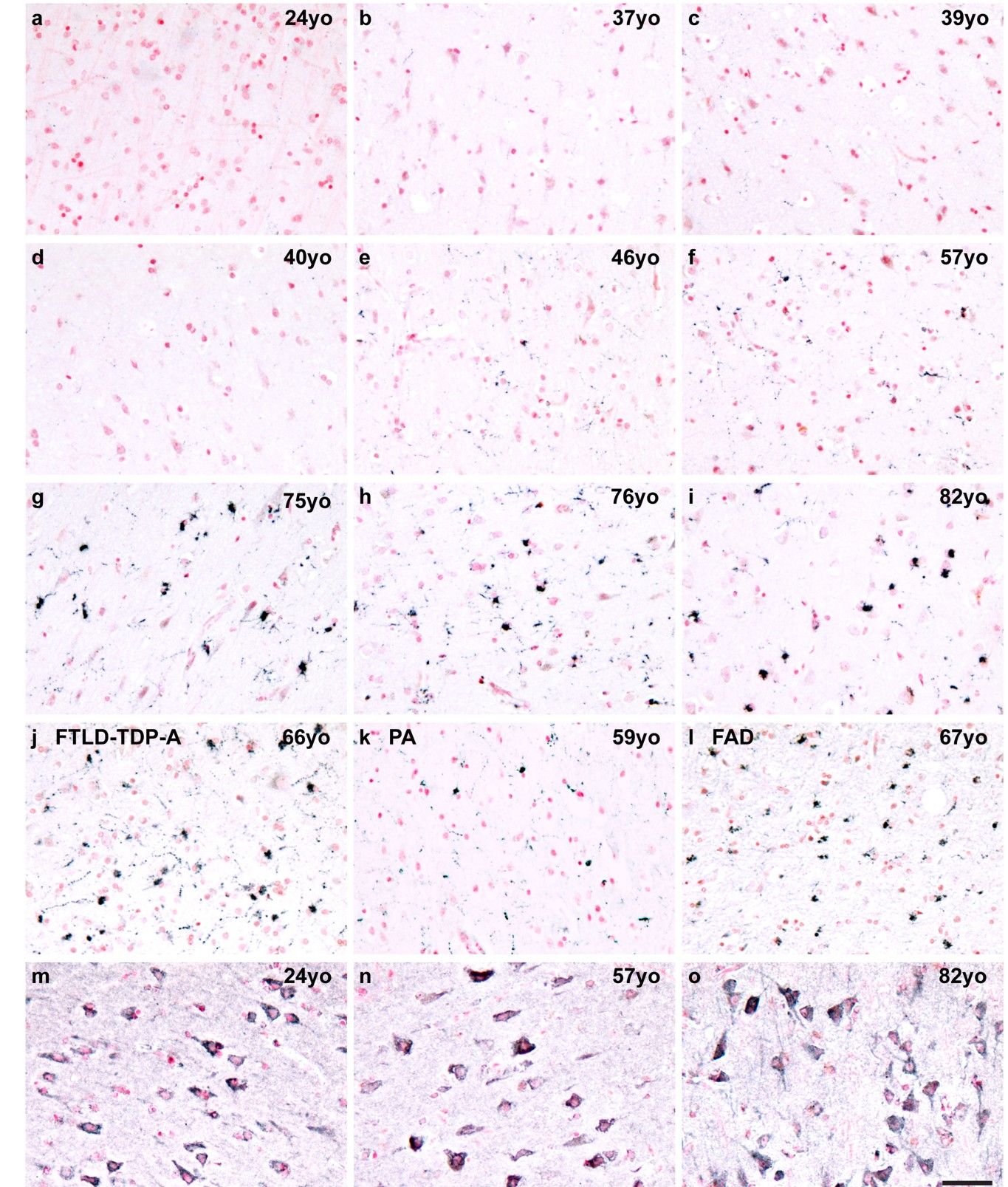

**Extended Data Fig. 8** | See next page for caption.

**Extended Data Fig. 8 | TMEM106B immunohistochemistry.** Immunostaining of sections from the frontal cortex of neurologically normal individuals aged 24–82 years (yo: year-old) (**a-i; m-o**) and individuals with abundant filamentous amyloid deposits made of various proteins (**j-l**). **a-l**. Sections were stained with C-terminal anti-TMEM239 antibody. **m-o**. Sections were stained with an N-terminal anti-TMEM106B antibody (Bethyl Labs). Nuclei were counterstained in red. Scale bar, 50 μm (**a-o**). TMEM106B inclusions were visualised with the C-terminal antibody, while the N-terminal antibody showed only diffuse cytoplasmic staining. Previous studies also reported diffuse cytoplasmic staining with N-terminal TMEM106B antibodies[29,41]. Staining with the C-terminal antibody showed age-dependent assembly of TMEM106B, with no inclusions in the frontal cortex of young individuals (**a-d**, and Fig. 3a), a small number in middle-aged individuals (**e, f**) and abundant inclusions in older individuals (**g-i** and Fig. 3b–e). Inclusions of TMEM106B were also present in cases with neurodegenerative diseases who were older than 59 years. By immunoblotting with antibody TMEM239, a 29 kDa band was present in cases with inclusions.

**Extended Data Table 1 | Further details on cases**

| Case | Disease | Gender | Mutation | Brain region | Publication |
|------|---------|--------|----------|--------------|-------------|
| 1 | AD | M | No | FL | Yang et al., 2022: sporadic AD case 1 |
| 2 | FAD | F | *PSEN1* (F105L) | FL | Yang et al., 2022: familial AD case 2 |
| 3 | EOAD | F | No | FL | - |
| 4 | PA | M | No | FL | Yang et al., 2022 |
| 5 | CBD | F | No | FL | Zhang et al., 2020: case 1 |
| 6 | CBD | F | No | FL | Zhang et al., 2020: case 2 |
| 7 | FTDP-17T | M | *MAPT* (P301L) | TL | - |
| 8 | AGD | M | No | NAcc | Shi et al., 2021: AGD case 1 |
| 9 | AGD | M | No | NAcc | Shi et al., 2021: AGD case 2 |
| 10 | LNT | F | No | FL | Shi et al., 2021 |
| 11 | ARTAG | F | No | HPC | Shi et al., 2021; Yang et al., 2022 |
| 12 | PD | M | No | CG | - |
| 13 | PDD | M | No | AMG | Yang et al., 2022 |
| 14 | FPD | | *SNCA* (G51D) | | - |
| 15 | DLB | M | No | FL | Schweighauser et al., 2020: DLB case 2 |
| 16 | DLB | M | No | FL | Yang et al., 2022 |
| 17 | MSA | F | No | PU | Schweighauser et al., 2020: MSA case 1 |
| 18 | MSA | M | No | PU | Schweighauser et al., 2020: MSA case 5 |
| 19 | MSA | F | No | PU | Schweighauser et al., 2020: MSA case 2 |
| 20 | FTLD-TDP-A | F | *GRN* (intronic) | FL | Yang et al., 2022 |
| 21 | FTLD-TDP-C | F | No | FL | - |
| 22 | ALS-TDP-B | F | No | MC | - |
| 23 | Control | M | No | FL | - |
| 24 | Control | M | No | FL | - |
| 25 | Control | M | No | FL | - |

AD: sporadic Alzheimer's disease; FAD: familial Alzheimer's disease; EOAD: sporadic early-onset Alzheimer's disease; PA: pathological aging; CBD: corticobasal degeneration; FTDP-17T: familial frontotemporal dementia and parkinsonism linked to chromosome 17 caused by *MAPT* mutations; AGD: argyrophilic grain disease; LNT: limbic-predominant neuronal inclusion body 4R tauopathy; ARTAG: aging-related tau astrogliopathy; PD: sporadic Parkinson's disease; PDD: sporadic Parkinson's disease dementia; FPD: familial Parkinson's disease; DLB: dementia with Lewy bodies; MSA: multiple system atrophy; FTLD-TDP-A: familial frontotemporal lobar degeneration with TDP-43 inclusions type A; FTLD-TDP-C: sporadic frontotemporal lobar degeneration with TDP-43 inclusions type C; ALS: amyotrophic lateral sclerosis; Control: neurologically normal individual. FL: frontal lobe; TL: temporal lobe; NAcc: nucleus accumbens; HPC: hippocampus; CG: cingulate gyrus; AMG: amygdala; PU: putamen; MC: motor cortex.

**Extended Data Table 2 | Cryo-EM data acquisition and structure determination**

| | Case 1 (AD) Fold I-s | Case 18 (MSA) Fold I-d | Case 19 (MSA) Fold IIa-s | Case 19 (MSA) Fold IIb-s | Case 17 (MSA) Fold III-s |
|---|---|---|---|---|---|
| **Data acquisition** | | | | | |
| Electron gun | CFEG | XFEG | XFEG | XFEG | XFEG |
| Detector | Falcon 4i | K2 | K2 | K2 | K2 |
| Energy filter slit (eV) | 10 | 20 | 20 | 20 | 20 |
| Magnification | 165,000 | 105,000 | 105,000 | 105,000 | 105,000 |
| Voltage (kV) | 300 | 300 | 300 | 300 | 300 |
| Electron dose (e–/Å$^2$) | 40 | 48.6 | 47.5 | 47.5 | 49.2 |
| Defocus range (µm) | 0.6 to 1.4 | 1.8 to 2.4 | 1.7 to 2.6 | 1.7 to 2.8 | 1.7 to 2.8 |
| Pixel size (Å) | 0.727 | 1.15 | 1.15 | 1.15 | 0.829 |
| | | | | | |
| **Map refinement** | | | | | |
| Symmetry imposed | C1 | C2 | C1 | C1 | C1 |
| Initial particle images (no.) | 47,414 | 17,027 | 67,797 | 67,797 | 121,045 |
| Final particle images (no.) | 47,414 | 12,057 | 7,347 | 12,855 | 24,142 |
| Map resolution (Å) | 2.64 | 3.64 | 3.38 | 3.47 | 2.76 |
| FSC threshold | 0.143 | 0.143 | 0.143 | 0.143 | 0.143 |
| Helical twist (°) | -0.42 | -0.41 | -0.63 | -0.49 | -0.69 |
| Helical rise (Å) | 4.81 | 4.78 | 4.78 | 4.79 | 4.79 |
| | | | | | |
| **Model Refinement** | | | | | |
| Model resolution (Å) | 2.6 | 3.74 | 3.22 | 3.33 | 2.41 |
| FSC threshold | 0.5 | 0.5 | 0.5 | 0.5 | 0.5 |
| Map sharpening $B$ factor (Å$^2$) | -37.57 | -45.37 | -29.29 | -34.66 | -43.93 |
| Model composition | | | | | |
| Non-hydrogen atoms | 3426 | 6852 | 3426 | 3426 | 3426 |
| Protein residues | 405 | 810 | 405 | 405 | 405 |
| Ligands | 12 | 24 | 12 | 12 | 12 |
| $B$ factors (Å$^2$) | | | | | |
| Protein | 43.78 | 43.85 | 43.85 | 43.85 | 44.64 |
| Ligand | 51.79 | 51.79 | 51.79 | 51.79 | 52.55 |
| R.m.s. deviations | | | | | |
| Bond lengths (Å) | 0.012 | 0.012 | 0.012 | 0.012 | 0.012 |
| Bond angles (°) | 2.181 | 2.189 | 2.222 | 2.380 | 2.024 |
| Validation | | | | | |
| MolProbity score | 0.84 | 0.90 | 1.16 | 1.77 | 1.05 |
| Clashscore | 0.00 | 0.15 | 0.31 | 0.59 | 0.15 |
| Poor rotamers (%) | 0.00 | 0.00 | 0.26 | 1.06 | 0.00 |
| Ramachandran plot | | | | | |
| Favored (%) | 94.24 | 93.98 | 89.97 | 88.22 | 91.98 |
| Allowed (%) | 5.76 | 6.02 | 10.03 | 11.78 | 8.02 |
| Disallowed (%) | 0.00 | 0.00 | 0.00 | 0.00 | 0.00 |

# nature research

# Reporting Summary

Nature Research wishes to improve the reproducibility of the work that we publish. This form provides structure for consistency and transparency in reporting. For further information on Nature Research policies, see Authors & Referees and the Editorial Policy Checklist.

## Statistics

For all statistical analyses, confirm that the following items are present in the figure legend, table legend, main text, or Methods section.

| n/a | Confirmed | |
|---|---|---|
| ☐ | ☒ | The exact sample size ($n$) for each experimental group/condition, given as a discrete number and unit of measurement |
| ☐ | ☒ | A statement on whether measurements were taken from distinct samples or whether the same sample was measured repeatedly |
| ☒ | ☐ | The statistical test(s) used AND whether they are one- or two-sided<br>*Only common tests should be described solely by name; describe more complex techniques in the Methods section.* |
| ☒ | ☐ | A description of all covariates tested |
| ☒ | ☐ | A description of any assumptions or corrections, such as tests of normality and adjustment for multiple comparisons |
| ☒ | ☐ | A full description of the statistical parameters including central tendency (e.g. means) or other basic estimates (e.g. regression coefficient) AND variation (e.g. standard deviation) or associated estimates of uncertainty (e.g. confidence intervals) |
| ☒ | ☐ | For null hypothesis testing, the test statistic (e.g. $F$, $t$, $r$) with confidence intervals, effect sizes, degrees of freedom and $P$ value noted<br>*Give P values as exact values whenever suitable.* |
| ☒ | ☐ | For Bayesian analysis, information on the choice of priors and Markov chain Monte Carlo settings |
| ☒ | ☐ | For hierarchical and complex designs, identification of the appropriate level for tests and full reporting of outcomes |
| ☒ | ☐ | Estimates of effect sizes (e.g. Cohen's $d$, Pearson's $r$), indicating how they were calculated |

*Our web collection on statistics for biologists contains articles on many of the points above.*

## Software and code

Policy information about availability of computer code

| Data collection | EPU (v1.11.1 and v2.3.079) |
|---|---|
| Data analysis | RELION (v3.1 and v4.0), CTFFIND (v4.1), COOT (v0.9), Chimera (v1.8.1), ChimeraX (v1.2), ISOLDE (v1.2), Adobe Photoshop 22.1.1, ImageJ (v2.1.0/1.53c) |

For manuscripts utilizing custom algorithms or software that are central to the research but not yet described in published literature, software must be made available to editors/reviewers. We strongly encourage code deposition in a community repository (e.g. GitHub). See the Nature Research guidelines for submitting code & software for further information.

## Data

Policy information about availability of data

All manuscripts must include a data availability statement. This statement should provide the following information, where applicable:
- Accession codes, unique identifiers, or web links for publicly available datasets
- A list of figures that have associated raw data
- A description of any restrictions on data availability

Raw cryo-EM micrographs are available in the Elecron Microscopy Public Image Archive (EMPIAR), entry numbers EMPIAR-10916 for case 1 (sporadic AD), EMPIAR-10968 for case 18 (MSA), EMPIAR-10358 for case 19 (MSA) and EMPIAR-10357 for case 17 (MSA). Cryo-EM maps have been deposited in the Electron Microscopy Data Bank (EMDB) under accession numbers EMD-14174 for I-s filaments of case 1, EMD-14176 for I-d filaments of case 18, EMD-14187 and EMD-14188 for IIa-s and IIb-s of case 19, respectively, and EMD-14189 for III-s of case 17. Refined atomic models have been deposited in the Protein Data Bank (PDB) under accession numbers 7QVC for I-s of case 1, 7QVF for I-d of case 18, 7QWG and 7QWL for IIa-s and IIb-s of case 19, respectively, and 7QWM for III-s of case 17.

# Field-specific reporting

Please select the one below that is the best fit for your research. If you are not sure, read the appropriate sections before making your selection.

☒ Life sciences ☐ Behavioural & social sciences ☐ Ecological, evolutionary & environmental sciences

For a reference copy of the document with all sections, see nature.com/documents/nr-reporting-summary-flat.pdf

# Life sciences study design

All studies must disclose on these points even when the disclosure is negative.

| | |
|---|---|
| Sample size | Frontal cortex from case 1 (sporadic AD), case 2 (inherited AD), case 3 (early-onset AD), case 4 (pathological ageing), cases 5 and 6 (both CBD), case 10 (LNT), cases 15 and 16 (both DLB), case 20 (FTLD-TDP-A), case 21 (FTLD-TDP-43), cases 23-25 and 12 additional cases (all neurologically normal individuals), temporal cortex from case 5 (FTDP-17T), case 14 (inherited PD), nucleus accumbens from cases 8 and 9 (both AGD), hippocampus from case 11 (ARTAG), cingulate cortex from case 12 (PD), amygdala from case 13 (PDD), putamen from cases 17, 18 and 19 (all MSA), and motor cortex from case 22 (ALS). All these samples were chosen based on availability of tissue (maximum available sample size). |
| Data exclusions | Pre-established common image classification procedures (S.H.W. Scheres, J. Struc. Biol. 180: 519-530, (2012)) were employed to select particle images with the highest resolution content in the cryo-EM reconstruction process. Details of the number of selected images are given in Extended Data Table 2. |
| Replication | All attempts at replication were successful. At least three independent biological repeats per experiment where representative data is shown. |
| Randomization | Randomisation was not performed, as it would not reduce any bias in this study, where samples are limited by brain availability. Therefore, samples were allocated in one experimental group (frontal cortex from cases 1-6, 10, 15, 16, 20, 21, 23-25, temporal cortex from cases 5 and 14, nucleus accumbens from cases 8 and 9, hippocampus from case 11, cingulate cortex from case 12, amygdala from case 13, putamen from cases 17-19, and motor cortex from case 22) based on neuropathological examination. |
| Blinding | Blinding was not performed, as the perceived risk of detection/performance bias was deemed negligible. |

# Reporting for specific materials, systems and methods

We require information from authors about some types of materials, experimental systems and methods used in many studies. Here, indicate whether each material, system or method listed is relevant to your study. If you are not sure if a list item applies to your research, read the appropriate section before selecting a response.

### Materials & experimental systems

| n/a | Involved in the study |
|---|---|
| ☐ | ☒ Antibodies |
| ☒ | ☐ Eukaryotic cell lines |
| ☒ | ☐ Palaeontology |
| ☒ | ☐ Animals and other organisms |
| ☐ | ☒ Human research participants |
| ☒ | ☐ Clinical data |

### Methods

| n/a | Involved in the study |
|---|---|
| ☒ | ☐ ChIP-seq |
| ☒ | ☐ Flow cytometry |
| ☒ | ☐ MRI-based neuroimaging |

## Antibodies

| | |
|---|---|
| Antibodies used | Primary antibody used for Western blotting was rabbit polyclonal TMEM239 used at 1:2,000. Primary antibodies used for immunohistochemistry were rabbit polyclonal TMEM239 used at 1:500 and rabbit polyclonal A303-439A (Bethyl Laboratories, catalogue number: A303-439A) used at 1:250. |
| Validation | TMEM239 validated against human TMEM106B residues 239-250 in Extended Data Figure 7. A303-439A was validated against recombinant human TMEM106B (Satoh, Ji. et al. Alz. Res. Therapy 6, 17 (2014). |

## Human research participants

Policy information about studies involving human research participants

| | |
|---|---|
| Population characteristics | See Table 1, Methods section, Extended Data Table 1, and Supplementary Table 1. Age at death: 79, 67, 58, 59, 74, 79, 55, 85, 90, 66, 85, 87, 64, 67, 74, 73, 85, 70, 68, 66, 65, 63, 75, 84, 101, 15, 20, 24, 25, 37, 39, 40, 46, 50, 57, 69, 76, 82; Gender: 16x female, 22x male; Diagnoses: 3x AD (sporadic, inherited, and early-onset), 1x PA, 2x CBD, 1x FTDP-17T, 2x AGD, 1x LNT, 1x |

ARTAG, 2x PD (sporadic and inherited), 1x PDD, 3x DLB (2x sporadic and early-onset), 3x MSA, 1x FTLD-TDP-A, 1x FTLD-TDP-C, 1x ALS, 15x neurologically normal individuals.

Recruitment

Samples were selected based on neuropathological examination and brain tissue availability, which is unlikely to have impacted the results.

Ethics oversight

The studies carried out at Indiana University, Tokyo Metropolitan Institute of Medical Science, UCL Queen Square Institute of Neurology, Edinburgh Brain and Tissue Bank, Toronto University, Vienna Medical University, Rotterdam University and Keio University were approved through the ethical review processes at each university's Institutional Review Board (IRB). Informed consent was obtained from the patients' next of kin.

Note that full information on the approval of the study protocol must also be provided in the manuscript.

