## [Peer Review File · Nature]

Manuscript Title: Age-dependent formation of TMEM106B amyloid filaments in human brains

Reviewer Comments & Author Rebuttals

Reviewer Reports on the Initial Version:

Referees' comments:

Referee #1 (Remarks to the Author):

Schweighauser et al. report the presence of TMEM106B fibrils comprised of peptide 120-254 in brain sarkosyl-insoluble brain extracts by cryo-EM. TMEM106B genetic variation has been associated with genetic risk for GRN-dependent FTLT-DTP, and to a lesser degree with other FTLT-DTP cases. As an endolysosomal protein, TMEM106B is thought to modulate TDP-43 pathology, though its mechanism is poorly defined. Previous studies have not documented TMEM106B aggregation, or co-aggregation with TDP-43. Here, the TMEM106B fibrils are observed in brain autopsy tissue extracts from multiple neurodegenerative diseases, and also in brain extracts from aged individuals without neurological disease. The cryo-EM identification of fibrils was correlated with the presence of a TMEM106B band immunoreactive with a C-terminal epitope in sarkosyl-insoluble extracts. The TMEM106B fibrils assume three different folds with all sharing the same structure through the first 47 aa residues, and containing either one or two protofilaments. The presence of TMEM106B fibrils appears to correlate most directly with age, having no obvious relationship to one or more neurodegenerative diseases. On the one hand, this suggests that TMEM106B fibrils may be a common and perhaps benign accompaniment of aging, similar to lipofuscin. On the other hand, TMEM106B is genetically linked to FTLT-DTP risk and amyloid fibril formation is rare, typically being associated with disease. This study is likely to attract high interest and stimulate further experimentation into the biological significance of TMEM106B fibrils. The data are convincing and clearly presented and the comments are minor.

1. The authors do not comment on T185S polymorphic residue linked with genetic risk. Is this variant predicted to alter fibril formation? Are the identified fibrils from individuals with zero, one or two variant alleles? This may bear on any possible biological significance of these fibrils.

2. The authors should be explicit as to why these fibers were not reported in previous studies of Tau and other fibers from brain extracts. The Methods suggest that the relative abundance of TMEM106B fibrils is strongly influenced by the timing of sarkosyl addition during the extraction. Explicit comments of this point would be helpful for other researchers.

3. The 29 kDa species identified in sarkosyl-insoluble extracts by anti-TMEM106B C-Terminal antibody is very likely the 120-254 fragment forming fibrils. The specificity should be bolstered by deglycosylation prior to SDS-PAGE of the extract to observe the predicted shift from 29 kDa to 17 kDa. Reactivity in the soluble fractions of brain should also be shown. Optimally, the antibody

specificity would be further validated by probing WT versus TMEM106B knockout mouse tissue.

4. Does the TMEM239 antibody detect TMEM106B aggregates in brain sections by IHC?

5. The authors have purified the recombinant protein TMEM106B (120-274). Some comment on its stability and solubility should be provided. Does the recombinant protein form amyloid-like aggregates in vitro under some experimental conditions?

6. The authors note that TMEM106B fibrils lack a “fuzzy coat”, whereas fibrils of tau, A β , α -synuclein and TDP-43 have such a surface. The authors might comment as to basis for this difference and the significance of a “coat” on other fibrils.

Referee #2 (Remarks to the Author):

Age-dependent formation of TMEM106B amyloid filaments in human brain
Schweighauser et al

The manuscript by Goedert and Scheres and colleagues recounts the next twist in the hugely popular, important and fascinating story of the amyloid fibril structures that are associated with different human diseases. The work in this manuscript builds on the beautiful previous studies from this group which have used cryo-EM to reveal a plethora of structures of Tau amyloid fibrils purified from the brains of individuals with different diseases, published recently in Nature, showing that a different fibril structure of the same, or very similar precursor sequence, is associated with a different disease. Their recent work on Abeta and its associated pathologies shows a similar story, although perhaps less striking than the observations with Tau, showing that different fibril structures are found in different diseases associated with Abeta aggregation.

Perhaps what is most interesting about neurodegeneration is that there is often no clear-cut diagnosis, with several diseases sharing overlapping features. Such disorders are often associated with the aggregation of more than one protein in each disease. The extraction procedures necessary for analysis of fibril structures using cryo-EM necessitates many steps of centrifugation and collection of sarkosyl-insoluble fractions wherein the amyloid fibrils reside. The consequence is that some fractions that are not analysed by cryoEM might contain crucial-disease associated materials and/or that molecules associated with fibrils might be removed from the fibril surfaces by the extraction procedure. Fine-tuning of the fibril-collection protocol could also reveal previously unseen fibril forms, as is the case described in the current manuscript. By including sarkosyl from the beginning of the fibril preparation new fibril forms have been discovered and, thanks to the high resolution of fibril structures now possible using cryoEM, shown here to be of a new protein, hitherto not known to form amyloid, residues 120-254 of the lysosomal protein, TMEM106B.

Using their now usual rigour, Goedert and Scheres collect fibril samples from 22 individuals with a wide range of amyloid disorders, spanning those associated with tau, TDP-43, Abeta, α -synuclein and show that these new TMEM106B fibrils are found in all of them. They are also found in ageing

individuals, without disease, but not in younger individuals, suggesting that the presence of TMEM106B fibrils themselves is not pathogenic, nor is their presence diagnostic of disease. This is an important finding, and raises the important point (that the authors could state more clearly) that not all fibrils are necessarily causative agents of neurodegeneration.

The current manuscript leaves many questions unanswered: what is the role of TMEM106 aggregation in disease: is it loss of function or gain of toxic function? This could be addressed, at least in an initial study, by expression of the C-terminal domain of TMEM106B in cells and assessment of cellular health. Given the central importance of lysosomes in cellular health and in proteostasis a phenotype might be expected. Does expression of the C-terminal TMEM106B domain result in a reduced ability of cells to degrade fibrils e.g. of tau or synculein? While it is clear that understanding the role of TMEM106B in human disease will require more detailed studies, including in animal models, cellular studies might at least cast some light on whether aggregation of TMEM106B affects lysosomal function or not.

Two other interesting points:

Polymorphism of residue 185 (Thr to Ser) has been associated with disease and T185 can be phosphorylated. At the resolution of the fibrils can the identity of residue 185 be discerned and it is post-translationally modified?

Given that TMEM106B is an immunoglobulin domain, the disulphide bond presumably links antiparallel beta strands in the native fold, but links parallel in-register strands in the amyloid fold. If so, global unfolding must be required for fibrils to form, as has been clearly shown by Fandrich et al. for LC amyloidosis. Some reference to this work would be helpful. Is the Ig domain of TMEM106B predicted to be particularly unstable (at lysosomal pH) and would that explain why it aggregates when released by proteolysis? Does its sequence have a high intrinsic amyloid potential? Please comment.

Overall, therefore, I found the article fascinating, and the fibril structures beautiful. The findings reported are important and will be of widespread interest. The work is perhaps a little premature for a journal such as Nature, since so many important and fundamental questions are left unanswered, such as the mechanism of aggregation, the identity of the protease causing cleavage, the effects on cellular function and the role in disease (loss of function/gain of toxic function/neither). However, given the novelty of these initial findings for so many diseases I would be excited to see the work reported in Nature.

Referee #3 (Remarks to the Author):

The manuscript by Schweighauser et al presents structures of TMEM106B fibrils, which were found in brain tissue from many patients with several different diseases and even in neurologically normal brains. This is an impressive body of work, which presents in total 28 cryo-EM structures from 24 different brain samples. The authors observed three slightly different folds, which, however, could not be clearly assigned to particular neurological disorders. The obtained results seem to provide

sufficient evidence that indicates that the presence of TMEM106B fibrils is not really related to the neurodegenerative diseases that the patients were suffering from, but is likely purely age-dependent. The presence (and of course also the structure) of these fibrils was so far unknown. This work is therefore an important further step in understanding not only the role of TMEM106B but also generally of amyloids in the brain.

The paper is very clearly written and the quality of the structural work is excellent. I do not see any issue that would have to be addressed and therefore recommend publication.

Author Rebuttals to Initial Comments:

Rebuttal

We thank the reviewers for their time and constructive comments, which have helped to improve the manuscript.

Referee #1 (Remarks to the Author):

Schweighauser et al. report the presence of TMEM106B fibrils comprised of peptide 120-254 in brain sarkosyl-insoluble brain extracts by cryo-EM. TMEM106B genetic variation has been associated with genetic risk for GRN-dependent FTLT-TDP, and to a lesser degree with other FTLT-TDP cases. As an endolysosomal protein, TMEM106B is thought to modulate TDP-43 pathology, though its mechanism is poorly defined. Previous studies have not documented TMEM106B aggregation, or co-aggregation with TDP-43. Here, the TMEM106B fibrils are observed in brain autopsy tissue extracts from multiple neurodegenerative diseases, and also in brain extracts from aged individuals without neurological disease. The cryo-EM identification of fibrils was correlated with the presence of a TMEM106B band immunoreactive with a C-terminal epitope in sarkosyl-insoluble extracts. The TMEM106B fibrils assume three different folds with all sharing the same structure through the first 47 aa residues, and containing either one or two protofilaments. The presence of TMEM106B fibrils appears to correlate most directly with age, having no obvious relationship to one or more neurodegenerative diseases. On the one hand, this suggests that TMEM106B fibrils may be a common and perhaps benign accompaniment of aging, similar to lipofuscin. On the other hand, TMEM106B is genetically linked to FTLT-TDP risk and amyloid fibril formation is rare, typically being associated with disease. This study is likely to attract high interest and stimulate further experimentation into the biological significance of TMEM106B fibrils. The data are convincing and clearly presented and the comments are minor.

1. The authors do not comment on T185S polymorphic residue linked with genetic risk. Is this variant predicted to alter fibril formation? Are the identified fibrils from individuals with zero, one or two variant alleles? This may bear on any possible biological significance of these fibrils.

We have now sequenced all cases described and added the results for the T185S polymorphism to Table 1. We discuss the revised text describing the significance of these findings reads:

“Genotyping of all cases (Table 1) showed that the alleles encoding T185 or S185 were equally represented. Individuals with fold I were homozygous for T185 or S185, or heterozygous, indicating that fold I can accommodate a threonine or a serine at position 185. Because of the compatibility of both residues with the glycosylation motif at N183, no differences in the associated glycan densities were observed. Fold II was only found in case 19, which was homozygous for T185. Seven out of eight cases with fold III were homozygous for S185, with the remaining case being heterozygous. It is possible that the packing of the side chain of residue 185 in the interior of fold III leaves insufficient space to accommodate a threonine.”

2. The authors should be explicit as to why these fibers were not reported in previous studies of Tau and other fibers from brain extracts. The Methods suggest that the relative abundance of TMEM106B fibrils is strongly influenced by the timing of sarkosyl addition during the extraction. Explicit comments of this point would be helpful for other researchers.

We have revised the Methods section, which now reads:

“The original sarkosyl extraction method, which we used in our work on the cryo-EM structures of tau filaments from Alzheimer’s disease, chronic traumatic encephalopathy and Pick’s disease (11-13), uses sarkosyl only after the first, low-speed centrifugation step (25). The method of Tarutani et al. (20) also uses sarkosyl at the beginning, i.e. before the first centrifugation step. This protocol change was essential for detecting abundant TMEM106B filaments, possibly because clumped filaments end up in the first pellet when sarkosyl is not yet present in the original method. In addition, the method of Tarutani et al uses a gentler clearing spin at the end, which results in an increase in the amount of filaments in the final sample.”

3. The 29 kDa species identified in sarkosyl-insoluble extracts by anti-TMEM106B C-terminal antibody is very likely the 120-254 fragment forming fibrils. The specificity should be bolstered by deglycosylation prior to SDS-PAGE of the extract to observe the predicted shift from 29 kDa to 17 kDa. Reactivity in the soluble fractions of brain should also be shown. Optimally, the antibody specificity would be further validated by probing WT versus TMEM106B knockout mouse tissue.

We performed deglycosylation using PNGase F. Following the prescribed procedures, we observed a large loss in the amount of detectable TMEM106B, using either TMEM239, or the commercially available AP22247b C-terminal antibody). Nevertheless, we did observe bands that were in agreement with the predicted shift from 29 kDa to 17 kDa upon deglycosylation (see immunoblots below; red arrows indicate the 17kDa bands). Because we do not understand the reasons for the loss in detectable material, we decided not to include this figure in the revised manuscript.

4. Does the TMEM239 antibody detect TMEM106B aggregates in brain sections by IHC?

Yes, it does. We have added Figure 3 and Extended Data Figure 8 to the manuscript. The revised text describing these experiments in the main text reads:

“In agreement with these observations, immunohistochemistry of brain sections with antibody TMEM239 showed staining of inclusions in disease cases and older control individuals, but not in younger controls (Figure 3; Extended Data Figure 8) It is not known how these inclusions relate to lysosomes.”

Previously, others published immunohistochemistry of brain sections with antibodies against the N-terminal part of TMEM106B, showing granular dot-like staining throughout nerve cells (27,39). We observed similar staining patterns with the antibody used in (27) (Extended Data Figure 8m-o) and note that the N-terminal epitopes for these antibodies are not present in the filaments.

5. The authors have purified the recombinant protein TMEM106B (120-274). Some comment on its stability and solubility should be provided. Does the recombinant protein form amyloid-like aggregates in vitro under some experimental conditions?

When expressed in *E.coli*, TMEM106B(120-274) is unstable and requires purification from inclusion bodies, as described in the Methods section. We have not attempted *in vitro* refolding of the material from inclusion bodies, or its assembly into filaments.

6. The authors note that TMEM106B fibrils lack a “fuzzy coat”, whereas fibrils of tau, A β , α -synuclein and TDP-43 have such a surface. The authors might comment as to basis for this difference and the significance of a “coat” on other fibrils.

This statement referred to the visual appearance of TMEM106B filaments in the cryo-EM micrographs, which was used to distinguish them from amyloids of other proteins. It is likely that the C-terminal 20 residues of TMEM106B do form a small fuzzy coat. To make this clearer, we modified the following sentences:

“... we observed a common type of filament that appeared to lack a fuzzy coat in the cryo-EM micrographs from cases of various conditions with abundant filamentous amyloid deposits”

and

“... they could be distinguished from filaments made of tau, A β , α -synuclein and TDP-43 by their general appearance and their apparent lack of a fuzzy coat”

Referee #2 (Remarks to the Author):

The manuscript by Goedert and Scheres and colleagues recounts the next twist in the hugely popular, important and fascinating story of the amyloid fibril structures that are associated with different human diseases. The work in this manuscript builds on the beautiful previous studies from this group which have used cryo-EM to reveal a plethora of structures of Tau amyloid fibrils purified from the brains of individuals with different diseases, published recently in Nature, showing that a different fibril structure of the same, or very similar precursor sequence, is associated with a different disease. Their recent work on Abeta and its associated pathologies shows a similar story, although perhaps less striking than the observations with Tau, showing that different fibril structures are found in different diseases associated with Abeta aggregation.

Perhaps what is most interesting about neurodegeneration is that there is often no clear-cut diagnosis, with several diseases sharing overlapping features. Such disorders are often associated with the aggregation of more than one protein in each disease. The extraction procedures necessary for analysis of fibril structures using cryo-EM necessitates many steps of centrifugation and collection of sarkosyl-insoluble fractions wherein the amyloid fibrils reside. The consequence is that some fractions that are not analysed by cryoEM might contain crucial-disease associated materials and/or that molecules associated with fibrils might be removed from the fibril surfaces by the extraction procedure. Fine-tuning of the fibril-collection protocol could also reveal previously unseen fibril forms, as is the case described in the current manuscript. By including sarkosyl from the beginning of the fibril preparation new fibril forms have been discovered and, thanks to the high resolution of fibril structures now possible using cryoEM, shown here to be of a new protein, hitherto not known to form amyloid, residues 120-254 of the lysosomal protein, TMEM106B.

Using their now usual rigour, Goedert and Scheres collect fibril samples from 22 individuals with a wide range of amyloid disorders, spanning those associated with tau, TDP-43, Abeta, α -synuclein and show that these new TMEM106B fibrils are found in all of them. They are also found in ageing individuals, without disease, but not in younger individuals, suggesting that the presence of TMEM106B fibrils themselves is not pathogenic, nor is their presence diagnostic of disease. This is an important

finding, and raises the important point (that the authors could state more clearly) that not all fibrils are necessarily causative agents of neurodegeneration.

The current manuscript leaves many questions unanswered: what is the role of TMEM106B aggregation in disease: is it loss of function or gain of toxic function? This could be addressed, at least in an initial study, by expression of the C-terminal domain of TMEM106B in cells and assessment of cellular health. Given the central importance of lysosomes in cellular health and in proteostasis a phenotype might be expected. Does expression of the C-terminal TMEM106B domain result in a reduced ability of cells to degrade fibrils e.g. of tau or synculein? While it is clear that understanding the role of TMEM106B in human disease will require more detailed studies, including in animal models, cellular studies might at least cast some light on whether aggregation of TMEM106B affects lysosomal function or not.

Our results suggest that TMEM106B aggregation does *not* affect neurological health, but rather that TMEM106B aggregation happens in an age-dependent manner. We thus feel that the question whether TMEM106B aggregation affects cellular health falls outside the scope of this work.

Two other interesting points:

Polymorphism of residue 185 (Thr to Ser) has been associated with disease and T185 can be phosphorylated. At the resolution of the fibrils can the identity of residue 185 be discerned and it is post-translationally modified?

The cryo-EM densities on their own do not unambiguously define the presence of an S or a T. However, we have now sequenced all cases and include the T185S polymorphism in Table 1. Considering these results, we have replaced T185 for a serine in Figure 1 panels b and d. Also see our answer to Referee #1 above for the changes in the main text.

Given that TMEM106B is an immunoglobulin domain, the disulphide bond presumably links antiparallel beta strands in the native fold, but links parallel in-register strands in the amyloid fold. If so, global unfolding must be required for fibrils to form, as has been clearly shown by Fandrich et al. for LC amyloidosis. Some reference to this work would be helpful. Is the Ig domain of TMEM106B predicted to be particularly unstable (at lysosomal pH) and would that explain why it aggregates when released by proteolysis? Does its sequence have a high intrinsic amyloid potential? Please comment.

The disulfide bond in the AlphaFold prediction of the globular fold of the C-terminal domain of TMEM106B links C214 at the end of an alpha-helix, with C253 in a loop. This sets TMEM106B apart from the LC amyloidosis case. We are not aware of predictions that indicate this Ig domain would be particularly unstable.

Overall, therefore, I found the article fascinating, and the fibril structures beautiful. The findings reported are important and will be of widespread interest. The work is perhaps a little premature for a journal such as Nature, since so many important and fundamental questions are left unanswered, such as the mechanism of aggregation, the identity of the protease causing cleavage, the effects on cellular function and the role in disease (loss of function/gain of toxic function/neither). However, given the

novelty of these initial findings for so many diseases I would be excited to see the work reported in Nature.

Referee #3 (Remarks to the Author):

The manuscript by Schweighauser et al presents structures of TMEM106B fibrils, which were found in brain tissue from many patients with several different diseases and even in neurologically normal brains. This is an impressive body of work, which presents in total 28 cryo-EM structures from 24 different brain samples. The authors observed three slightly different folds, which, however, could not be clearly assigned to particular neurological disorders. The obtained results seem to provide sufficient evidence that indicates that the presence of TMEM106B fibrils is not really related to the neurodegenerative diseases that the patients were suffering from, but is likely purely age-dependent. The presence (and of course also the structure) of these fibrils was so far unknown. This work is therefore an important further step in understanding not only the role of TMEM106B but also generally of amyloids in the brain.

The paper is very clearly written and the quality of the structural work is excellent. I do not see any issue that would have to be addressed and therefore recommend publication.

Reviewer Reports on the First Revision:

Referees' comments:

Referee #1 (Remarks to the Author):

The authors have substantially improved the manuscript and done an excellent job of addressing questions from the first round of review. There are two minor comments they might consider though I would not consider them essential.

1. The new immunohistochemistry with anti-TMEM239 antibody in Fig 3 and Ext Data Fig 8 are an important addition. Given this appearance, is there a colocalization with age-dependent lipofuscin accumulation, as detected by simple autofluorescence? This might be assessed in the same or immediately adjacent paraffin sections. Perhaps TMEM106B fibrils are a component of lipofuscin.
2. The authors tried the PNGase deglycosylation experiment and included in the rebuttal. Given the loss of immunoreactivity, the decision not to include in the manuscript makes sense. On a minor note, it would still be helpful to include a blot of sarkosyl-soluble fractions to show that intact TMEM106B protein is present in all samples.

Referee #2 (Remarks to the Author):

I have read the revised version of this manuscript and the responses to the three reviewers. Genotyping the individuals so as to discern whether there is a Ser or Thr at position 185 is an important and informative addition. The immunohistochemistry added with the anti-TMEM106B antibodies in the main text and Extended data is also a welcome addition. It was disappointing that the authors were not willing to characterise the effects of expressing the C-terminal domain of TMEM106B in cells. I was also disappointed that the authors did not add a comment on the stability, pH stability and predicted amyloidogenicity of TMEM106B. This would be an insightful addition especially given amyloid formation is not related to disease. Nonetheless, this is another stunning piece of work from the Goedert and Scheres team that will excite the readers of Nature and the field.

Author Rebuttals to First Revision:

Responses to remaining referees' comments:

Referee #1 (Remarks to the Author):

The authors have substantially improved the manuscript and done an excellent job of addressing questions from the first round of review. There are two minor comments they might consider though I would not consider them essential.

1. The new immunohistochemistry with anti-TMEM239 antibody in Fig 3 and Ext Data Fig 8 are an important addition. Given this appearance, is there a colocalization with age-dependent lipofuscin accumulation, as detected by simple autofluorescence? This might be assessed in the same or immediately adjacent paraffin sections. Perhaps TMEM106B fibrils are a component of lipofuscin.

We have modified the following sentence to the revised manuscript: "Like lipofuscin, a lysosomal complex of oxidised proteins and lipids that develops in an age-dependent manner in many tissues (25), TMEM106B filaments may also form in lysosomes, even though staining for TMEM106B inclusions was not always associated with the presence of lipofuscin autofluorescence"

2. The authors tried the PNGase deglycosylation experiment and included in the rebuttal. Given the loss of immunoreactivity, the decision not to include in the manuscript makes sense. On a minor note, it would still be helpful to include a blot of sarkosyl-soluble fractions to show that intact TMEM106B protein is present in all samples.

Antibody TMEM239 does not see full-length TMEM106B by immunoblotting. We showed this using HEK293T cells expressing HA-tagged TMEM106B.

Referee #2 (Remarks to the Author):

I have read the revised version of this manuscript and the responses to the three reviewers. Genotyping the individuals so as to discern whether there is a Ser or Thr at position 185 is an important and informative addition. The immunohistochemistry added with the anti-TMEM106B antibodies in the main text and Extended data is also a welcome addition. It was disappointing that the authors were not willing to characterise the effects of expressing the C-terminal domain of TMEM106B in cells. I was also disappointed that the authors did not add a comment on the stability, pH stability and predicted amyloidogenicity of TMEM106B. This would be an insightful addition especially given amyloid formation is not related to disease. Nonetheless, this is another stunning piece of work from the Goedert and Scheres team that will excite the readers of Nature and the field.

We respectfully disagree with the referee and thank them for their support.